# Smoothed monthly Greenland ice sheet elevation changes during 2003-2023

Shfaqat A. Khan*[1], Helene Seroussi[2], Mathieu Morlighem[3], William Colgan[4], Veit Helm[5], Gong Cheng[3], Danjal Berg[1], Valentina R. Barletta[1], Nicolaj K. Larsen[6], William Kochtitzky[7], Michiel van den Broeke[8], Kurt H. Kjær[6], Andy Aschwanden[9], Brice Noël[10], Jason E. Box[4], Joseph A. MacGregor[11], Robert S. Fausto[4], Kenneth D. Mankoff[12,13], Ian M. Howat[14], Kuba Oniszk[1], Dominik Fahrner[4], Anja Løkkegaard[4], Eigil Y. H. Lippert[1], Alicia Bråtner[1], Javed Hassan[1]

[1]DTU Space, Technical University of Denmark, Denmark
[2]Thayer School of Engineering, Dartmouth College, Hanover, NH, USA
[3]Department of Earth Sciences, Dartmouth College, Hanover, NH, USA
[4]Department of Glaciology and Climate, Geological Survey of Denmark and Greenland, Denmark
[5]Alfred-Wegener-Institut Helmholtz-Zentrum für Polar- und Meeresforschung, Bremerhaven, Germany
[6]Globe Institute, University of Copenhagen, 1350 Copenhagen, Denmark
[7]School of Marine and Environmental Programs, University of New England, Biddeford, ME, USA
[8]Institute for Marine and Atmospheric Research, Utrecht University, The Netherlands
[9]University of Alaska Fairbanks, Fairbanks, AK, USA
[10]Laboratoire de Climatologie et Topoclimatologie, University of Liège, Liège, Belgium.
[11]Cryospheric Sciences Lab, NASA Goddard Space Flight Center, Greenbelt, MD, USA
[12]NASA Goddard Institute for Space Studies, New York, NY, 10025 USA
[13]Autonomic Integra LLC, New York, NY, 10025 USA
[14]Byrd Polar Research Center and School of Earth Sciences, Ohio State University, Columbus, OH, USA

Correspondence to: Shfaqat Abbas Khan (abbas@space.dtu.dk)

**Abstract.** The surface elevation of the Greenland Ice Sheet is constantly changing due to the interplay between surface mass balance processes and ice dynamics, each exhibiting distinct spatiotemporal patterns. Here, we employ satellite and airborne altimetry data with fine spatial (1 km) and temporal (monthly) resolutions to document this spatiotemporal evolution from January 2003 to August 2023. To estimate elevation changes of the Greenland Ice Sheet (GIS), we utilize radar altimetry data from CryoSat-2 and EnviSat, laser altimetry data from the ICESat and ICESat-2, and laser altimetry data from NASA's Operation IceBridge Airborne Topographic Mapper. We produce continuous monthly ice surface elevation changes from January 2003 to August 2023 on a 1 km grid covering the entire GIS. We estimate cumulative ice loss of 4,352 Gt ± 315 Gt (12.1 ± 0.9 mm sea level equivalent) during this period, excluding peripheral glaciers. Between 2003 and 2023, the ice sheet land-terminating margin underwent a significant cumulative thinning of several meters. Ocean-terminating glaciers exhibited thinning between 20–40 m, with Jakobshavn Isbræ experiencing an exceptional thinning of nearly 70 m. This dataset of fine-

resolution altimetry data in both space and time will support studies of ice mass loss and useful for GIS ice sheet modelling.

To validate our monthly mass changes of the Greenland ice sheet, we use mass change from satellite gravimetry and mass change from the Input-Output method. On multiannual timescales, there is a strong correlation between the time series, with R values ranging from 0.88 to 0.92.

## 1 Introduction

Over the last three decades, satellite-based observations have revealed unprecedented details regarding the Greenland Ice
Sheet's (GIS) mass balance and its response to a warming climate. This wealth of satellite data has not only allowed for the quantification of mass loss, but also offered insights into the complex interactions between atmospheric, oceanic, and glaciological processes influencing the ice sheet's response *(Box et al.*, 2022; *Khan et al.*, 2022a; *Sasgen et al.*, 2020; *van den Broeke et al.*, 2016; *Wood et al.*, 2021). The Gravity Recovery and Climate Experiment (GRACE) and GRACE Follow-On (FO) missions have played a crucial role in this endeavor, revealing a significant ice loss of 4,550 ± 784 gigatons (Gt)
(equivalent to 12.6 ± 2.2 mm of sea level equivalent, SLE) during the period from 2002 to 2019 *(Velicogna et al.*, 2020). The input–output method (IOM), which generates the longest continuous time series of mass change among the most commonly used methods, indicates an accelerated mass loss for the GIS over the past four decades *(Mouginot et al.*, 2019; *Mankoff et al., 2021*). Notably, IOM permits separation of the total mass loss into its component processes, which indicates that ice discharge remained relatively constant from 1972 to 2002 and thereafter escalated due to the acceleration of multiple outlet
glaciers.

Satellite and airborne altimetry, on the other hand, present a direct measurement approach for tracking changes in the ice sheet thickness, expressed as changes in surface elevation at a finer spatial resolution of a few kilometers. Previous altimetry studies have relied on observations from a single satellite mission or the fusion of multi-sensor data to estimate trends over varying time intervals, typically ranging between 1 to 10 years *(Bamber and Dawson*, 2020; *Csatho et al.*, 2014; *Gardner et*
*al.*, 2013; *Helm et al.*, 2014; *Hurkmans et al.*, 2014; *Schenk et al.*, 2014; *Shepherd et al.*, 2020; *Benjamin Smith et al.*, 2021; *Benjamin Smith et al.*, 2019; *Sørensen et al.*, 2018; Winstrup et al., 2024). Few studies have produced sub-annual elevation change estimates. Smith et al. (2023) used ICESat-2 to measure the net surface-height change of the GIS at 3-month resolution to validate surface-height differences predicted by three combinations of climate- and firn-densification models. *Slater et al.* (2021) used CryoSat-2 satellite altimetry during 2011-2020 to produce direct measurements of Greenland's
runoff variability, based on seasonal changes in the ice sheet's surface elevation. However, they relied on average values over the entire ablation zone. *Lai and L. Wang* (2021) estimated GIS surface elevation changes with a 30-day resolution and a 5.5 km × 5.5 km spatial resolution using altimetry data from ICESat-1, CryoSat-2, and ICESat-2 from 2003 to 2020. Their approach allows for the integration of surface elevations measured by multiple missions by estimating a mission-dependent bias parameter. *Ravinder et al.* (2024) used CryoSat-2 and ICESat-2 data to estimate seasonal and interannual elevation

changes and showed good agreement between CryoSat-2 and ICESat-2, with the best agreement occurring in North Greenland, where the measurements are relatively dense.

Typically, the spatial and temporal resolution of elevation change products is constrained by the resolution of satellite ground tracks. Satellite altimetry generally provides 2-5 repeat measurements per year over the same location. To achieve

higher temporal resolution, such as monthly measurements, elevation changes must be averaged over a large area, resulting in coarse spatial resolution. Alternatively, combining observations with a model representing ice surface changes can achieve both high spatial and temporal resolution. This study employs the latter approach. We enhance the method from Khan et al. (2022a) to generate continuous monthly surface elevation changes from 2003 to 2023 on a 1 km grid covering the entire Greenland Ice Sheet. This new dataset, alongside other recent sub-annual elevation change estimates, forms a basis for

detecting and understanding short-term mass loss fluctuations on both local and regional scales and their impact on long-term trends.

## 2. Input Data

### 2.1 Radar and laser altimetry data

To estimate the monthly mass changes of the Greenland Ice Sheet (GIS) from January 2003 to August 2023, we utilize the

following datasets: (1) radar altimetry data from ESA's CryoSat-2 mission (Wingham et al., 2006), (2) radar altimetry data from ESA's Environmental Satellite (EnviSat) (Rémy et al., 2015), (3) laser altimetry data from the Ice, Cloud, and land Elevation Satellite (ICESat) (Zwally et al., 2014), (4) laser altimetry data from the Ice, Cloud, and land Elevation Satellite-2 (ICESat-2) (Smith et al., 2021), and (5) laser altimetry data from NASA's Operation IceBridge Airborne Topographic Mapper (ATM) flights (Studinger, 2020) (see Figure 1).

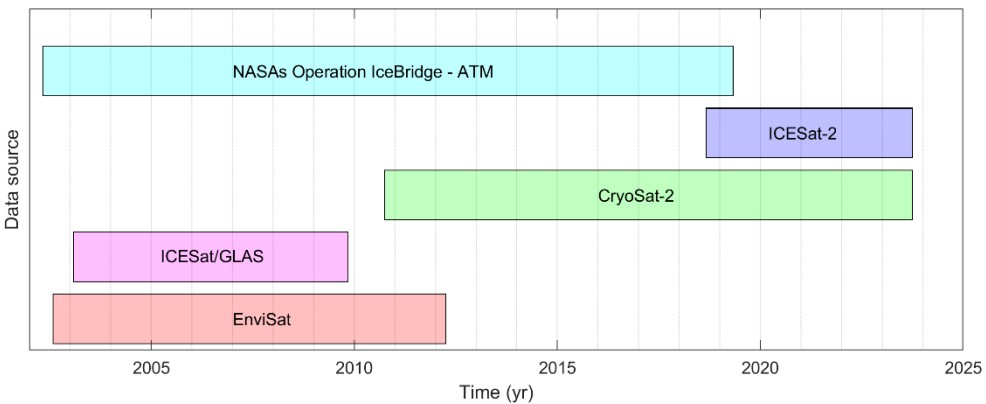


**Figure 1. Temporal coverage of the airborne and satellite altimetry missions used to estimate monthly elevation changes.**

**CryoSat-2**: We incorporate all available CryoSat-2 data covering the Greenland ice sheet from November 2010 to August 2023. Data processing follows Khan et al. (2022), utilizing overlapping ground tracks to generate surface elevation time series. Points with series shorter than three years were excluded. The spatial coverage of CryoSat-2 elevation time series is illustrated in Figure 2, showing sparse data in regions with steep slopes, such as the central west Greenland terminus of Jakobshavn Isbræ (Figure 2c). Overall, we use approximately 1.024 billion single point measurements to create about 10.5 million point time series.

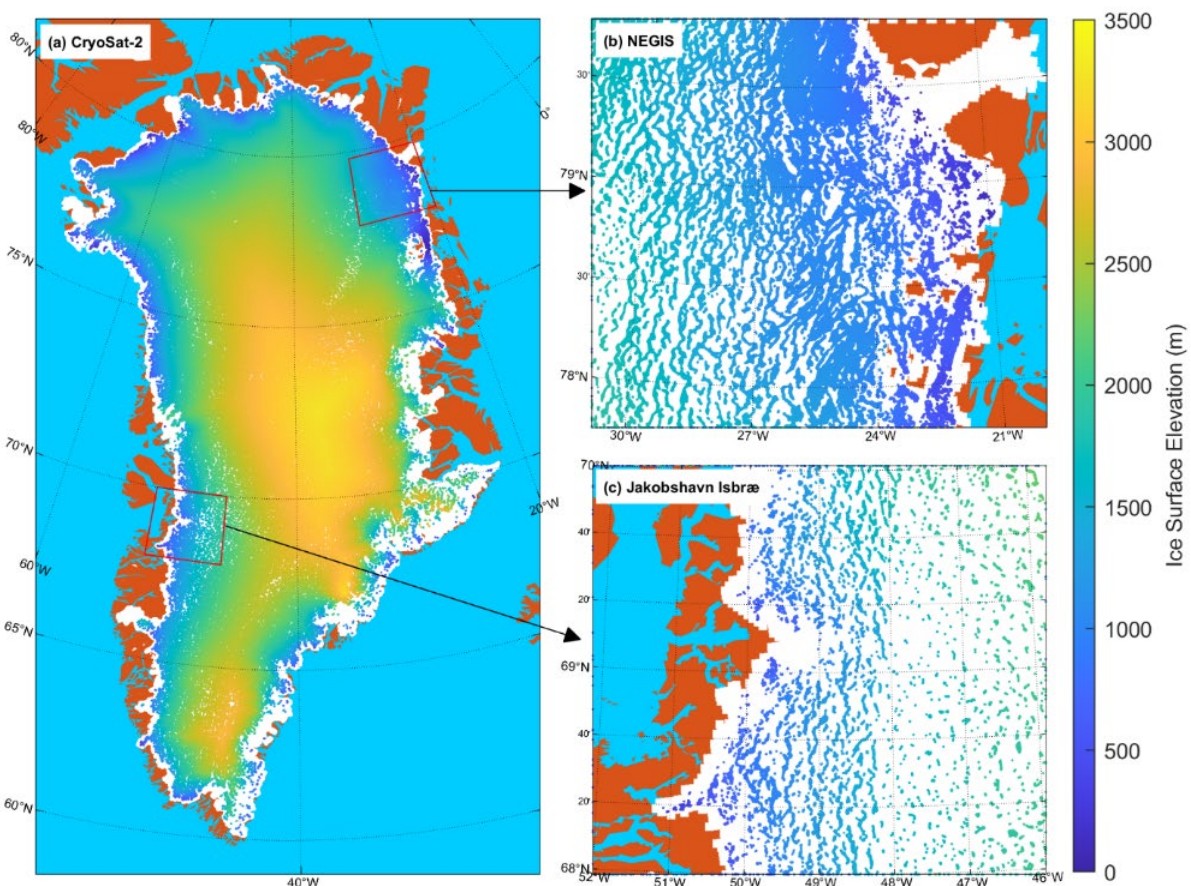

**Figure 2. (a) Spatial coverage of points with elevation time series from CryoSat-2. (b) Elevation time series in northeast Greenland. (c) Elevation time series at Jakobshavn Isbræ. The colorbar denotes ice surface elevation.**

**EnviSat:** We use all available EnviSat data from August 2002 to March 2012, processed similarly to CryoSat-2 data. Overlapping ground tracks are employed to create surface elevation time series, excluding points with series shorter than

three years. In total, we utilize about 328.6 million single point measurements to generate approximately 2.0 million point time series (figure 3a).

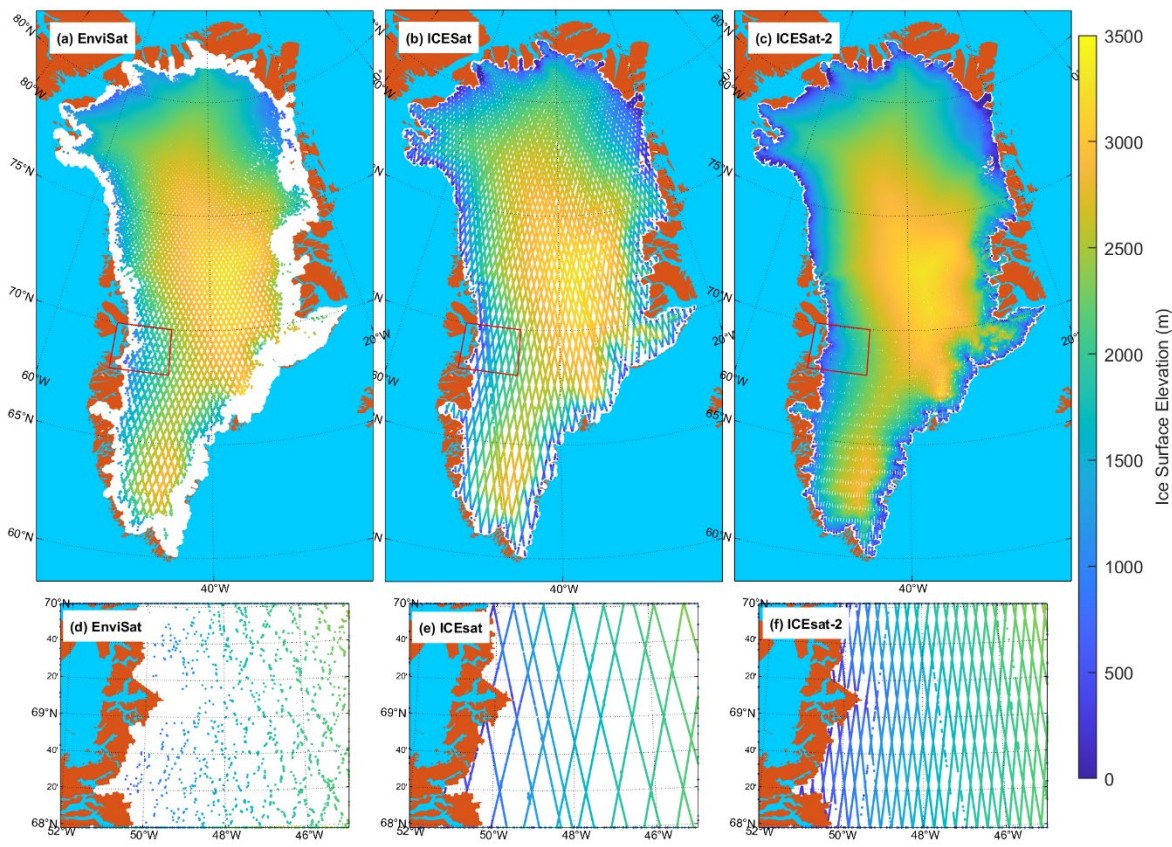

**Figure 3. Spatial coverage of points with elevation time series from (a) EnviSat, (b) ICESat, and (c) ICESat-2.**

**ICESat:** We include all available ICESat data from February 2003 to September 2009 (Schenk and Csatho, 2012; Smith et al., 2020; Zwally et al., 2014), specifically using GLAS/ICESat L2 Global Antarctic and Greenland Ice Sheet Altimetry Data (HDF5), Version 34 (Zwally, 2014). Points with overlapping ground tracks are used to create surface elevation time series, excluding those shorter than three years. This results in the use of about 303.8 million single point measurements to produce approximately 3.4 million point time series. Figure 3b shows the coverage of ICESat point time series.

**ICESat-2**: All available ICESat-2 data from November 2018 to August 2023 is utilized, specifically ATLAS/ICESat-2 L3A Land Ice Height, Version 6 (Smith et al., 2023). Following the same method as ICESat and CryoSat-2, we use overlapping ground tracks to create surface elevation time series, excluding points with series shorter than three years. This results in about 3.209 billion single point measurements to create approximately 6.1 million point time series. Figure 3 illustrates the

spatial coverage of ICESat, ICESat-2, and EnviSat data, showing that laser altimetry (ICESat, ICESat-2) better covers the ice sheet margins compared to radar altimetry (EnviSat, CryoSat-2).


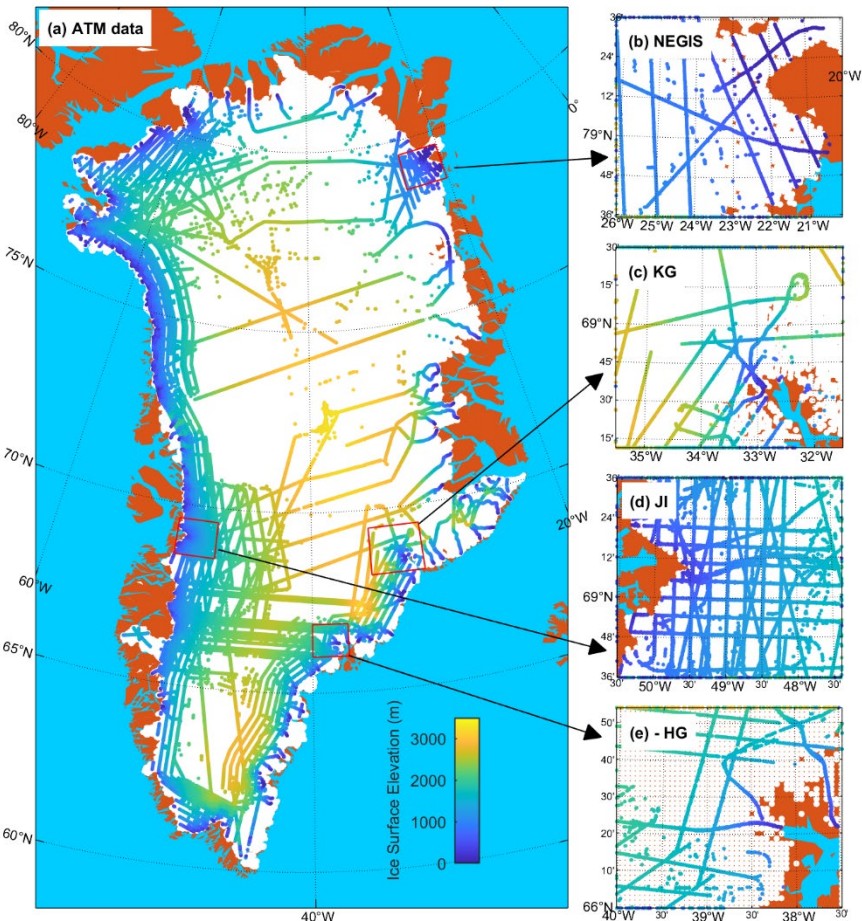

**Figure 4. (a) Spatial coverage of points with elevation time series from NASA's ATM flight. Elevation time series at northeast Greenland (b), Kangerlussuaq Glacier (c), Jakobshavn Isbræ (d), and Helheim Glacier (e).**

**NASA's ATM flights:** To enhance data coverage near the ice margin, we supplement satellite altimetry with airborne altimetry. We use all available laser altimetry data from NASA's Operation IceBridge ATM flights from April 2002 to April 2019 (Studinger, 2020), utilizing overlapping ground tracks to create surface elevation time series and excluding those shorter than three years. This results in about 175.1 million single point measurements to create approximately 0.7 million point time series. Figure 4a shows the coverage of ATM point time series. Notably, several main outlet glaciers, including

the main flowlines of Helheim Glacier (HG), Jakobshavn Isbræ (JI), Kangerlussuaq Glacier (KG), and the northeast Greenland Ice Stream (NEGIS), were repeatedly overflown during 2003-2019. Characteristics of different sensors used in this study is shown in Table 1.


**Table 1: Characteristics of the different sensors used in this study.**

| Sensor | Time span | Footprint | Single point accuracy | Citation |
|---|---|---|---|---|
| ICESat | Feb. 2003 to Sep. 2009 | 70 m | ± 15 cm | Zwally et al., 2014; Schutz et al., 2005 |
| ICESat-2 | Nov. 2018 to Aug 2023 | 13 m | ± 4 cm | Neumann et al., 2019; Markus et al., 2017 |
| EnviSat | Aug. 2002 to Mar. 2012 | 5 – 15 km | ± 10 – 15 cm | Benveniste et al., 2002; ESA, 2010 |
| CryoSat-2 | Nov. 2010 to Aug. 2023 | LRM (1.5 km) SAR and SARIn Mode (250 m along-track and 1.5 km across-track) | ± 10 – 15 cm | Gourmelen et al., 2018; Wingham et al., 2006 |
| ATM flights | April 2002 to April 2019 | 1 – 5 m depending on flight altitude | ± 5 – 10 cm | Studinger, 2020; Miles et al., 2013 |

## 3. Methods

### 3.1 Improvement compared to previous study


Here, we improve the method used by *Khan et al.* (2022a) and provide continuous monthly surface elevation changes during 2003-2023 on a 1x1 km grid covering the entire ice sheet. While *Khan et al.* (2022a) estimated elevation changes over a ten-year period (2011–2022) using annual mean elevation changes, our approach extends the temporal coverage to twenty years (2003–2023) and introduces monthly mean elevation changes. This expanded temporal range and improved monthly

resolution allow for a more detailed capture of fine-scale elevation dynamics. Additionally, this study incorporates ICESat and Envisat data, enhancing the comprehensiveness of the observational dataset. Here, we apply a revised seasonal function to more accurately capture the unequal distribution of surface thickening and thinning over the year, with approximately eight months of surface elevation increases and four months of decreases, closely aligning with observed seasonal trends.

Together with recent sub-annual elevation change estimates, this dataset provides an essential foundation for detecting and understanding short-term mass loss fluctuations on local and regional scales, as well as their influence on long-term trends.

## 3.2 Monthly elevation changes from ICESat

We use ICESat to estimate monthly elevation changes from February 2003 to September 2009 *(Zwally et al.*, 2014). To estimate elevation changes over the ice sheet we follow the procedure described by *Khan et al.* (2022a). We employ a regular grid with a 1x1 km resolution that spans the entire GIS. The center of each grid point is denoted as $\mathbf{C}(x_0, y_0)$. For every grid point, we select all ICESat data at coordinates $\mathbf{P}(x_i, y_{i,}, h_i, t_i)$, within 1000 m of $\mathbf{C}$, where $\mathbf{P}$ includes elevation values $h_i$ measured at time $t_i$. The index $i$ denotes each specific data point.

Utilizing all available ICESat data collected between February 2003 to September 2009 (Zwally et al., 2014), we generate surface elevation time series at each grid point $\mathbf{C}$. To depict surface changes, we employ a 7th-order polynomial to characterize temporal elevation changes and a 3rd-order polynomial equation to describe the surface shape. Additionally, a seasonal term is incorporated to address seasonal surface variations. For each grid point with the center at $(x_0, y_0)$, we identify the nearest data point within a 1000 m radius $(x_i, y_i, h_i, t_i)$ and apply the 7th-order polynomial $H(t_i)_{poly}$, the 3rd-order surface topography $H_{topo}$, and the seasonal term $H(t_i)_{seasonal}$:

$$H(t_i) = H(t_i)_{poly} + H_{topo} + H(t_i)_{seasonal} \quad (1)$$

The 7th-order polynomial is:

$$H(t_i)_{poly} = a_1 + a_2 t_i + a_3 t_i^2 + a_4 t_i^3 + a_5 t_i^4 + a_6 t_i^5 + a_7 t_i^6 + a_8 t_i^7 , \quad (2)$$

where $t_i$ is the time when the *i*-th measurement was observed. For simplicity, we used 1 January 2010 as reference time t=0. $a_1$ to $a_8$ are polynomial variables.

To describe the surface, we fit a 3rd-order polynomial to the observed elevations over the area of 1x1 km (Csatho et al., 2014):

$$H_{topo} = a_9 x + a_{10} y + a_{11} x^2 + a_{12} y^2 + a_{13} xy + a_{14} x^3 + a_{15} y^3 + a_{16} xy^2 + a_{17} x^2 y, \quad (3)$$

where $a_9$ to $a_{17}$ are parameters that describe the slope and concavity/convexity of the surface, $x$ and $y$ are coordinates of the ICESat data point, but in a system with $x_0$ and $y_0$ as the center, i.e., $x = x_i - x_o$ and $y = y_i - y_o$. The seasonal term is given by:

$$H(t_i)_{s1} = a_{18} cos(\omega_1 t_i + a_{19}) + 0.5 a_{18} cos(\omega_2 t_i + 2a_{19} + 0.5\pi), \quad (4)$$

where $a_{18}$ denotes the amplitude, $a_{19}$ is the phase, and $\omega_1$ and $\omega_2$ are the frequencies of the annual (1 yr) and the semi-annual (0.5 yr) term, respectively.

Most previous studies employ a single cosine function to describe seasonal surface elevation changes at any point on the ice sheet:

$$H(t_i)_{s2} = A_i cos(\omega_1 t_i + \varphi_i), \quad (5)$$

Here, we use the following function in equation 4 to better capture the unequal distribution of surface thickening and thinning over the year. The two functions, $H(t_i)_{s1}$ and $H(t_i)_{s2}$, are displayed in Figure 5.

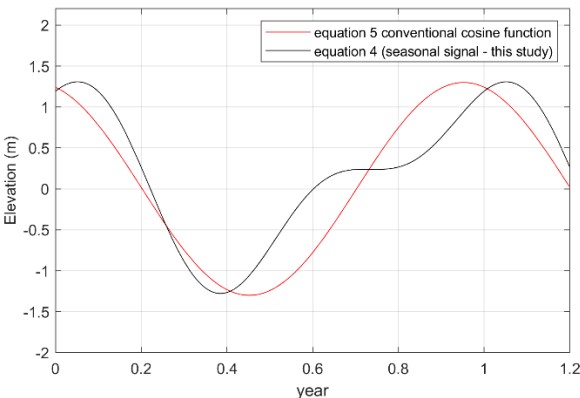

**Figure 5. Seasonal evolution with the black curve showing the seasonal signal used in this study and the red curve showing the**
**cosine function used in previous studies.**

The Surface Mass Balance (SMB) and GRACE time series suggest ice mass gain over 8 months, from approximately mid-August to mid-April, and rapid mass loss during the remaining 4 months (see discussion). To describe this behaviour, we combine annual and semi-annual functions. However, we predefine the parameters of the semi-annual term, so that the function reproduces surface elevation increases for approximately 8 months and surface lowering for 4 months. In principle,
fitting an annual and semi-annual signal to surface elevations would require estimating four unknown parameters. However, the temporal resolution of satellite altimetry data is often poor and does not strongly constrain all parameters. For instance, ICESat had a repeat track of 3 months, but many points have only 2-3 observations per year, preventing the separation of the annual and semi-annual signals over the entire ice sheet. Using the above equation, we only need to estimate two unknowns, $a_{18}$ and $a_{19}$.

For each grid point, we create a time series and use least squares adjustment to simultaneously estimate parameters $a_1$ to $a_{19}$.

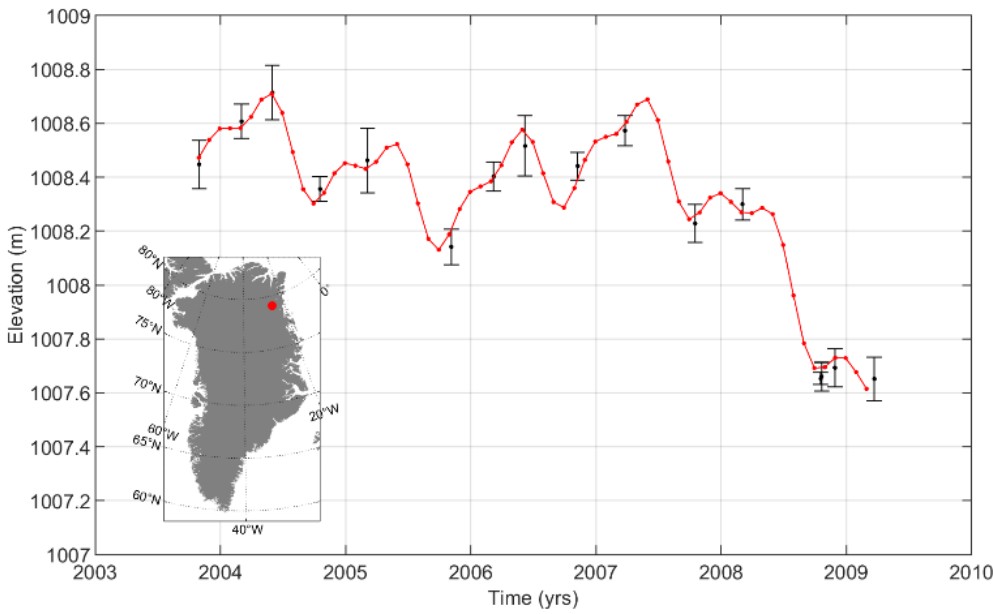

**Figure 6. Surface elevation change time series derived from ICESsat data for a single point. The location of the point is shown as a red dot on the map of Greenland. The solid red curves show the best-fitting 7th-order polynomial and the seasonal signal corrected for surface topography. The error bars denote observed elevations.**


Figure 6 displays a time series of surface elevation corrected for 3rd-order surface topography. The red curve shows the best-fitting 7th-order polynomial + the seasonal term, $H(t_i)_{s1}$. We fit a polynomial only if the observed time series has a length of >3 yrs. Furthermore, we detect and remove outliers from each time series. We define an outlier as a point outside $2\sigma$ (standard deviation) of the residual signal (difference between observed elevation and predicted elevation from the

polynomial fit). We use the parameters $a_1$ to $a_{19}$ for each grid point to estimate elevation changes over the entire GIS for consecutive 1-month periods. Spatial coverage of ICESat data is shown in Figure 3b.

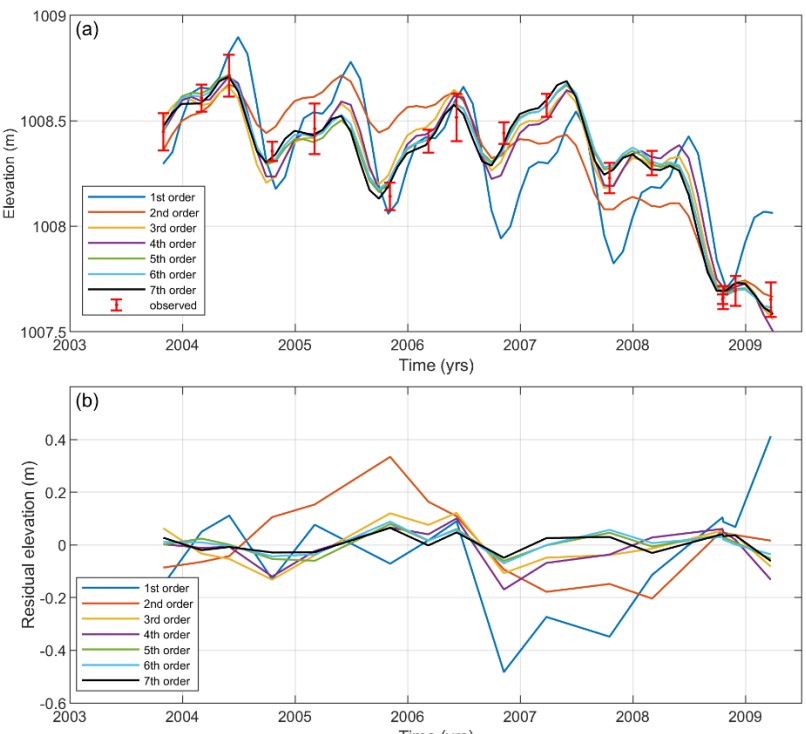

Figure 7. (a) Surface elevation change time series derived from ICESsat data for same point as in figure 6. The solid curves show the best-fitting polynomial or order 1 to 7, and the seasonal signal. (b) differences between the observed elevations and the polynomial fits.

In Figure 7a, we present the best-fitting polynomials of orders 1 through 7, along with the seasonal component. The residuals (figure 7b), defined as the difference between the observed elevation and the polynomial fit, indicate that for polynomials of orders 5 to 7, the residuals are approximately ±10 cm.

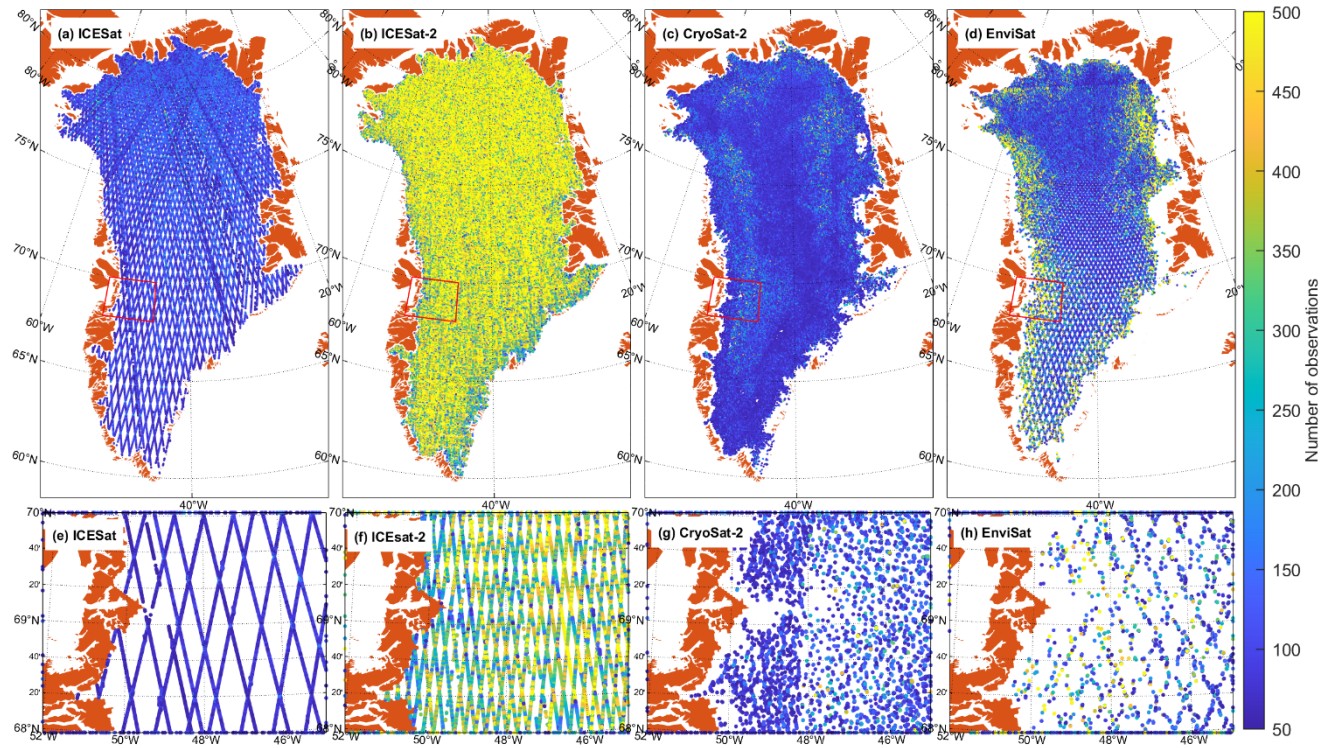

**Figure 8. Total number of observations used per grid point for (a) ICESat, (b) ICESat-2, (c) CryoSat-2, and (d) EnviSat. The lower panels show the same as top panel but for the Jakobshavn Isbræ region, (e) ICESat, (f) ICESat-2, (g) CryoSat-2, and (h) EnviSat.**

Figure 8a and 8e show the total number of observations used per grid point for ICESat. A threshold of a minimum of 50 observations is applied, excluding any time series with fewer data points. To estimate parameters, we incorporate all observations within a 1 km radius of the center grid point. This ensures a sufficient number of observations to reliably estimate all parameters (in total 19), including the 7th-order polynomial, 3rd-order surface topography, and seasonal term.

As shown in Figure 7, polynomials of order 5–7 effectively represent these changes. However, polynomial selection is constrained by data availability. To estimate parameters, we incorporate all observations within a 1 km radius of the center grid point. While a 500 m radius could be used, it would lead to large areas with insufficient observations and potential overfitting issues. We use a 3rd-order polynomial to represent surface topography, however, the choice of polynomial order dependent on the selected radius. A larger radius (e.g., 5 km) requires a higher-order polynomial to capture complex topographic variations, whereas a smaller radius (e.g., 500 m) allows for a simpler 1st- or 2nd-order polynomial. Our selection of polynomials for describing both surface changes and topography is a balance between ensuring sufficient observations and reliably estimating all parameters (19 in total). To assess parameter reliability, we display the RMS of residuals (figure 9) from point time series for each sensor. Notably, RMS values are highest near the margin, where surface

topography is more complex and may require higher-order polynomials. Alternatively, integrating high-resolution (10×10 m) Digital Elevation Model (DEM) data could further improve topographic representation.

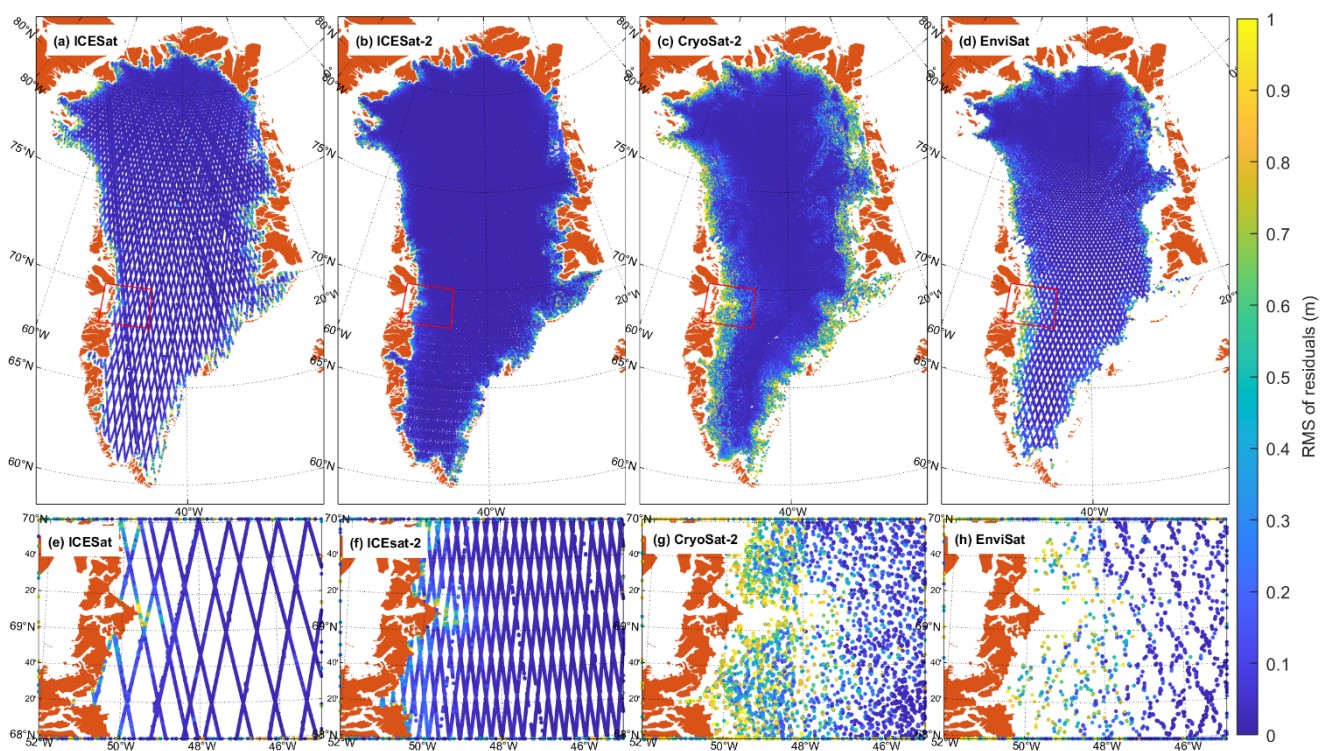

Figure 9. RMS of residuals for (a) ICESat, (b) ICESat-2, (c) CryoSat-2, and (d) EnviSat. The lower panels show the same as top panel but for the Jakobshavn Isbræ region, (e) ICESat, (f) ICESat-2, (g) CryoSat-2, and (h) EnviSat.

**ICESat-2:** Our method of deriving monthly surface elevation changes from ICESat-2 is identical to ICESat.

### 3.3 Monthly elevation changes from CryoSat-2

Our procedure for processing CryoSat-2 is identical to *Khan et al.* (2022a). To estimate elevation changes using CryoSat-2 data, we follow the procedure of ICESat and ICESat-2 data, however, with a minor modification regarding the seasonal signal, $H(t_i)_{s1}$. We employ a regular grid with a 1x1 km resolution that spans the entire GIS. Utilizing all available CryoSat-2 data collected between November 2010 to October 2023, we generate surface elevation time series at each grid point.

For each grid point, we use point time series to estimate a 7th-order polynomial $H(t_i)_{poly}$, a 3rd-order surface topography $H_{topo}$, and a seasonal term $H(t_i)_{seasonal}$

Earlier studies (and this study for ICESat and ICESat-2) assume the shape of the surface remains constant (Schenk et al., 2014) throughout the studied period. However, near the ice margin, the shape of the surface may change significantly over the course of 20 years. As a compromise, rather than fitting a polynomial to the entire 2010-2023 period, we consider two individual time sub-intervals separately, the first one between 2010-2017, and the second one between 2017-2023. During each sub-interval we assume the shape of the surface remains constant. This allows the shape of the surface to change over longer intervals.

270

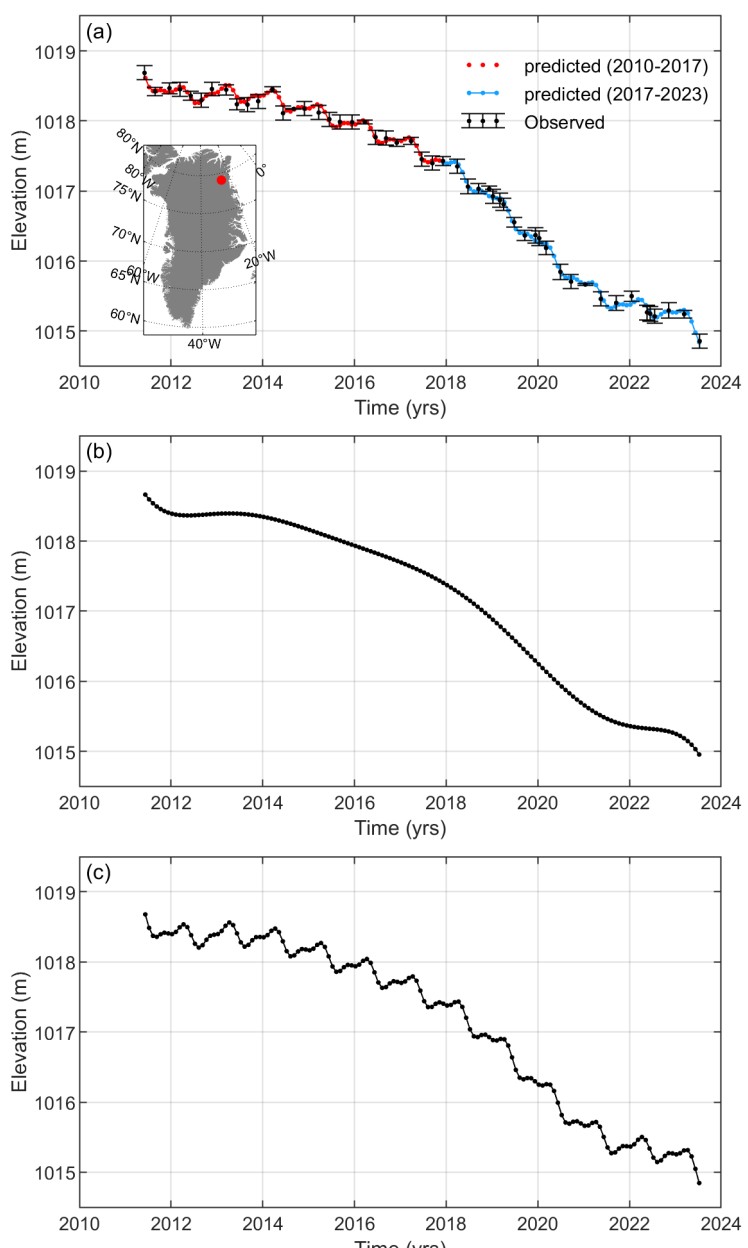

**Figure 10. Surface elevation change time series derived from Cryosat-2 data for a single point. The location of the point is shown as a red dot on the map of Greenland. (a) The solid curves show the best-fitting 7th-order polynomial and the seasonal signal corrected for surface topography. The error bars denote observed elevations. (b) The solid curves show the best-fitting 7th-order polynomial corrected for the seasonal signal and surface topography. (c) The solid curves show the best-fitting 7th-order polynomial corrected for surface topography, however, a seasonal signal from ICESat/ICESat-2 is added.**

Figure 10a displays a time series of surface elevation corrected for $3^{rd}$-order surface topography. The red and blue curve shows the best-fitting 7th-order polynomial + the seasonal term. We fit a polynomial only if the observed time series has a length of >3 yrs. Furthermore, we detect and remove outliers from each time series. Outliers are identified based on residuals, which represent the difference between the observed elevation and the polynomial fit. Any values falling outside the 2-σ range are excluded. We use the parameters $a_1$ to $a_{19}$ for each grid point and each (sub-)interval, e.g., 2010-2017 and 2017-2023, to estimate elevation changes over the entire GIS for consecutive 1-month periods. Spatial coverage of CryoSat-2 data is shown in Figure 2.

**CryoSat-2 seasonal signal**

The radar signal from CryoSat-2 might not be reflected by the snow surface but could instead penetrate through dry snow. There may be minor local discrepancies that may be attributed to the penetration of the radar signal into the snow. To address this issue, we remove a seasonal signal estimated from CryoSat-2, $H(t_i)_{seasonal,cryo}$, from point time series, and substitute it with a seasonal signal, $H(t_i)_{seasonal,ice1/ice2}$, estimated from ICESat and ICESat-2. The stepwise procedure is illustrated in Figure 11 and depicted for a point time series in Figure 10.

Estimate CryoSat-2 time series at point $i$ using:
7th-order polynomial
3rd-order surface topography
$H_{topo}$ Seasonal term $H(t_i)_{seasonal}$

$$H(t_i) = H(t_i)_{poly} + H_{topo} + H(t_i)_{seasonal,cryo}$$

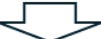

Remove Seasonal term $H(t_i)_{seasonal,cryo}$ estimated from CryoSat-2 data

$$H(t_i) = H(t_i)_{poly} + H_{topo}$$

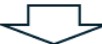

Add Seasonal term $H(t_i)_{seasonal,ice1/ice2}$ estimated from ICEsat and ICEsat-2 data

$$H(t_i) = H(t_i)_{poly} + H_{topo} + H(t_i)_{seasonal,ice/ice2}$$

**Figure 11. The stepwise procedure on how a seasonal signal from CryoSat-2 data is replaced with a seasonal signal from ICESat and ICESat-2.**

In this study, we employ ICESat and ICESat-2 to obtain the seasonal signal between 2003-2009 and 2018-2023, respectively. For the 2009-2018 period, , we derive the mean seasonal signal from ICESat and ICESat-2 (mean of $a_{18}$ and $a_{19}$ from ICESat and ICESat-2 seasonal signal). We propose that the mean amplitude from ICESat and ICESat-2 serves as a

reasonable approximation for filling the gap from 2009 to 2018 (see section 3.5).

### 3.4 Monthly elevation changes from EnviSat

To estimate elevation changes using EnviSat data, we follow the procedure of CryoSat-2 data. We employ a regular grid with a 1x1 km resolution that spans the entire GIS. Utilizing all available EnviSat data collected between August 2002 to

March 2012, we generate surface elevation time series at each grid point. For each grid point, we use point time series to estimate a 7th-order polynomial $H(t_i)_{poly}$, a 3rd-order surface topography $H_{topo}$, and a seasonal term $H(t_i)_{seasonal,Envi}$

Next, we remove a seasonal signal estimated from EnviSat, $H(t_i)_{seasonal,envi}$, from point time series, and substitute it with a seasonal signal, $H(t_i)_{seasonal,ice1/ice2}$, estimated from ICESat and ICESat-2. Figure 3a shows spatial coverage of EnviSat time series.

## 3.5 Amplitude of the Seasonal signal

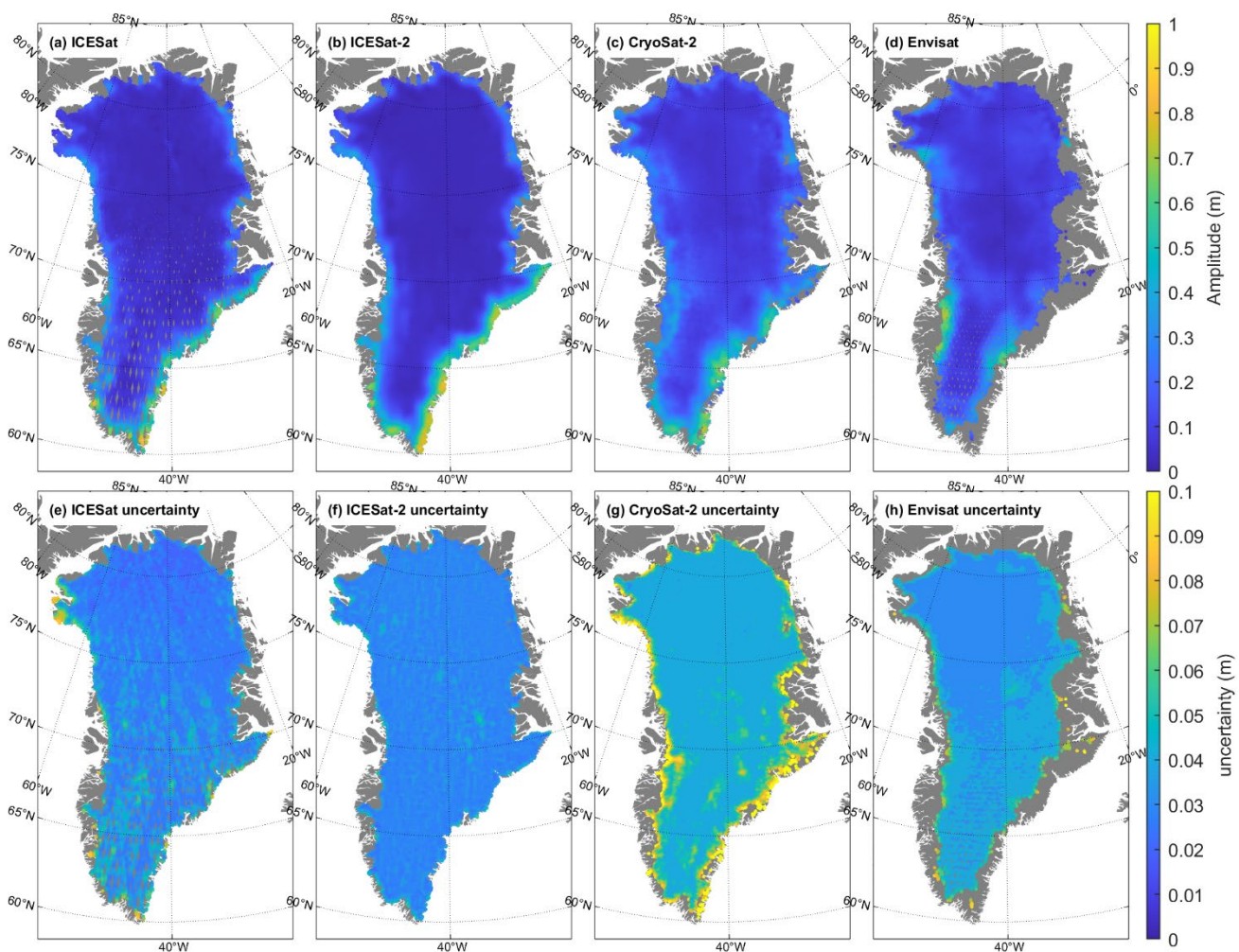

**Figure 12. Amplitude of seasonal surface elevation changes from (a) ICESat, (b) ICESat-2 (c) CryoSat-2, and (d) EnviSat.**

The seasonal signal amplitudes derived from ICESat, ICESat-2, CryoSat-2, and EnviSat satellite missions, are shown in
Figure 12. Predominantly, the highest amplitudes are observed in southeast Greenland across all satellite missions. It is important to note that EnviSat data is not inclusive of the ice sheet margin. The slight variations in amplitudes can be attributed, in part, to the differing time spans utilized for amplitude estimation.

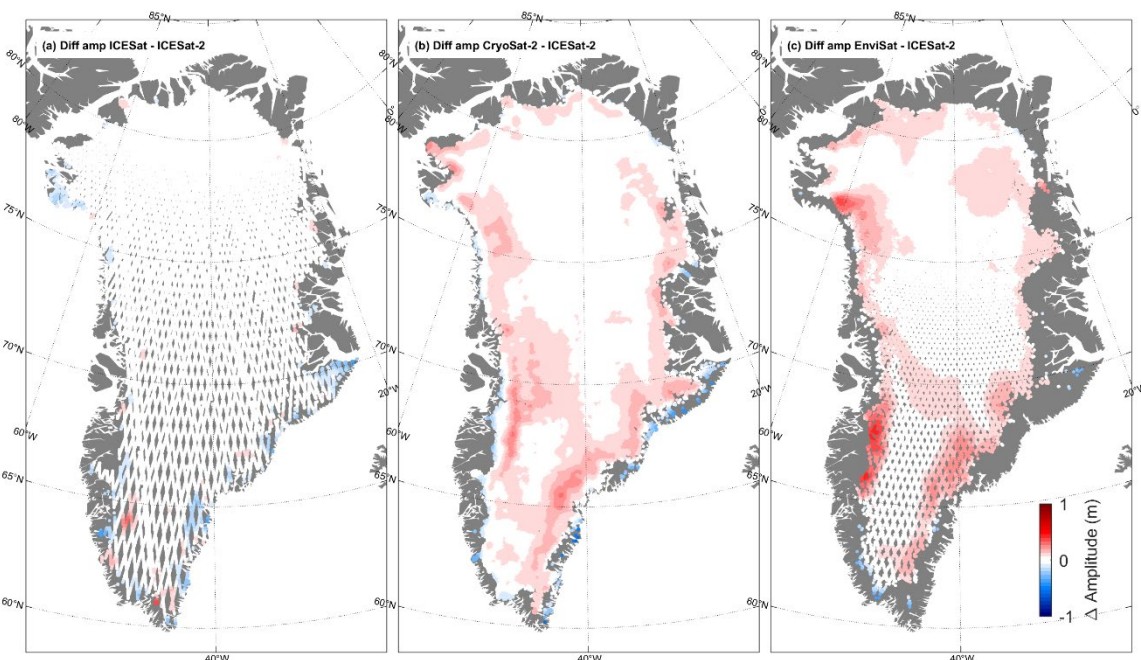

**Figure 13. Difference in seasonal amplitude between (a) ICESat-2 and ICESat, (b) ICESat-2 and CryoSat-2, (c) ICESat-2 and EnviSat.**

Figure 13 show the difference in seasonal amplitude between ICESat-2 and ICESat, CryoSat-2, EnviSat, respectively. Notably, the amplitude difference between ICESat and ICESat-2 is small. This discrepancy may stem from the fact that amplitudes are estimated over different time periods using data from two different sensors with varying spatial and temporal resolutions. Given the strong overall agreement between ICESat and ICESat-2, we propose that the mean amplitude from ICESat and ICESat-2 serves as a reasonable approximation for filling the gap from 2009 to 2018. Figure 13b and 13c suggest that Envisat and Cryosat-2 yields larger amplitude compared to icesat-2. Therefore, in this study, we use seasonal signal from ICESat and ICESat-2. For the 2009-2018 period, we derive mean seasonal signal from ICESat and ICESat-2. Figure 12(e-f) show standard deviation of the amplitude at each grid point, $i$, from ICESat and ICESat-2, $\sigma_{i,icesat,amp}$ and $\sigma_{i,icesat2,amp}$, respectively. For the 2009-2018 period, we use standard error based on mean ICESat and ICESat-2 amplitudes.

The seasonal layer is the top layer on the ice sheet. For this layer we use a constant density of $315\pm44$ kg/m$^3$ to convert snow volume to mass (Fausto at al., 2018).

### 3.6 Monthly elevation changes from NASA's Operation IceBridge ATM flights

During 2002-2019, NASA conducted annual airborne surveys with ATM over the GIS during the spring. These flights were mostly concentrated along the margins of the GIS. To estimate monthly changes, we use the same approach as used for satellite altimetry data described above. However, spring data alone do not allow us to extract a seasonal signal. Therefore, $H(t_i)_{seasonal}$ is not estimated for ATM data. Instead, we adopt a seasonal signal estimated from ICESat and ICESat-2 data. In addition, to allow the shape of the surface to change, we divide the study period into two separate (sub-)intervals, i.e., 2002-2010 and 2010-2019. During each (sub-) interval we assume the shape of the surface remains constant. Figure 4 shows Spatial coverage of the ATM time series.

### 3.7 Monthly elevation changes and their uncertainty

For each grid point, we use point time series to estimate 7th-order polynomial, a 3rd-order surface topography, and a seasonal term from a synthesis of several sensors' datasets is somewhat different from other studies *(Nilsson et al.*, 2022; *Simonsen et al.*, 2021). While other studies typically estimate the bias between sensors/missions, we do not merge data from different satellite missions and therefore do not estimate any biases. Instead, we estimate monthly elevation changes for each sensor's dataset independently as described in previous sections, and only merge the estimated monthly change rates from each dataset afterwards, when creating the multi-sensor monthly grid. Figure 14 illustrates this process for June 2008, for which we merged monthly elevation changes from NASA's ATM data, ICESat data and Envisat data. To detect local outliers of monthly rates, we used planar regression in 20 km bins. Values that fell outside the 5-σ range are excluded. The estimated monthly elevation changes were used to interpolate elevation change rates onto a regular grid of 1x1 km. For each grid point, we use ordinary kriging to interpolate elevation change rates $dh_{i,krig}$ and the associated error $\sigma_{i,krig}$ *(Khan et al.*, 2022a).

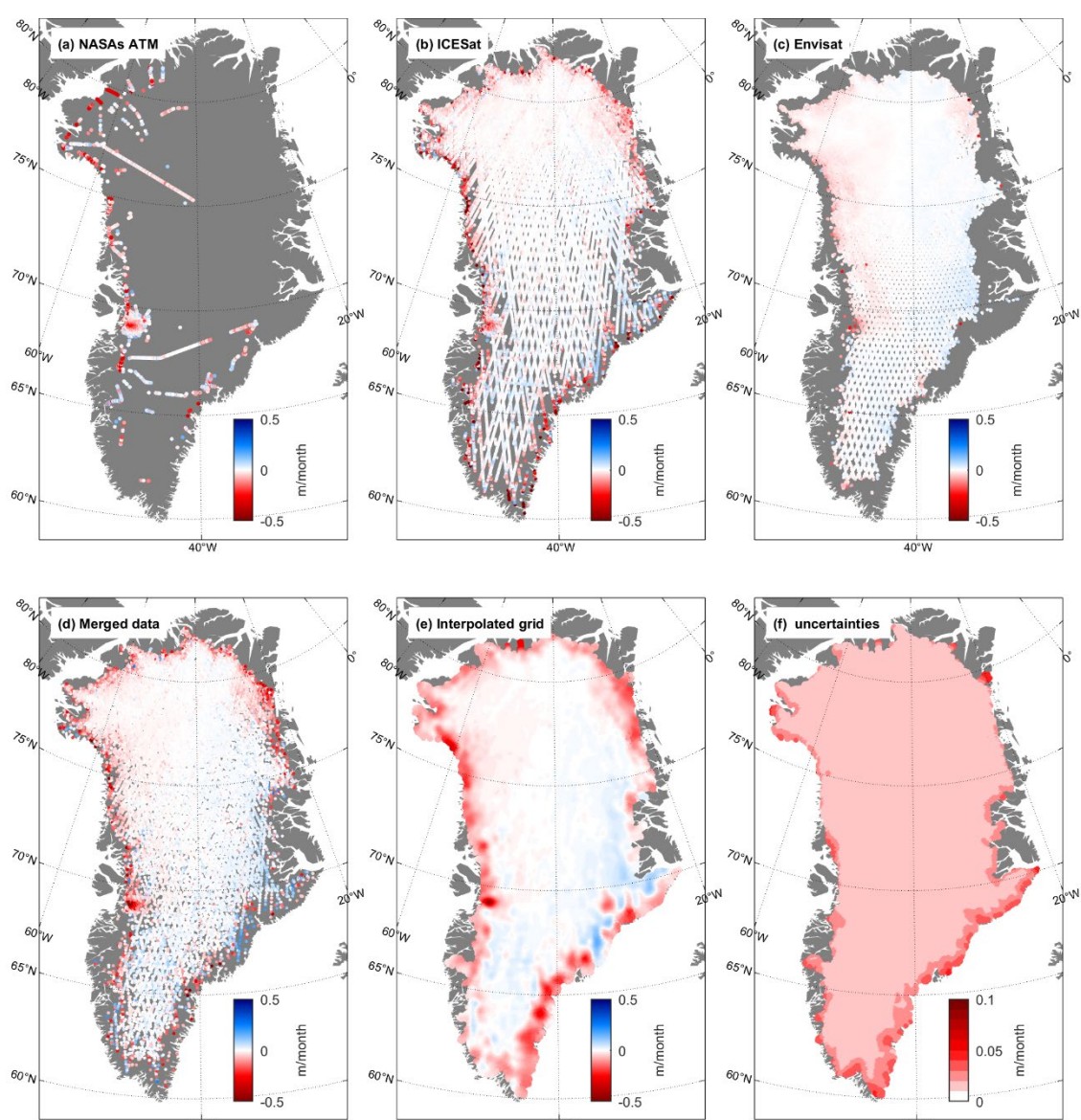

**Figure 14. Surface elevation change during June 2008 from (a) ATM flights, (b) ICESat laser satellite altimetry, (c) EnviSat radar altimetry, (d) combined elevation changes from airborne and satellite data, (e) interpolated elevation changes onto a 1x1 km grid, and (f) uncertainties of the interpolated data.**

Figure 14b shows a much denser distribution of monthly elevation change points in northern Greenland compared to the south. However, the number of points with monthly elevation changes varies over time. In the 1×1 km grid interpolation using kriging shown in Figure 14e, the average percentage of effective raw grids—representing the area covered by data points—was about 10%. Figure 15 presents the average percentage of effective raw grids for each month from 2003 to 2023, with the best coverage observed when both ICESat-2 and CryoSat-2 data are available.

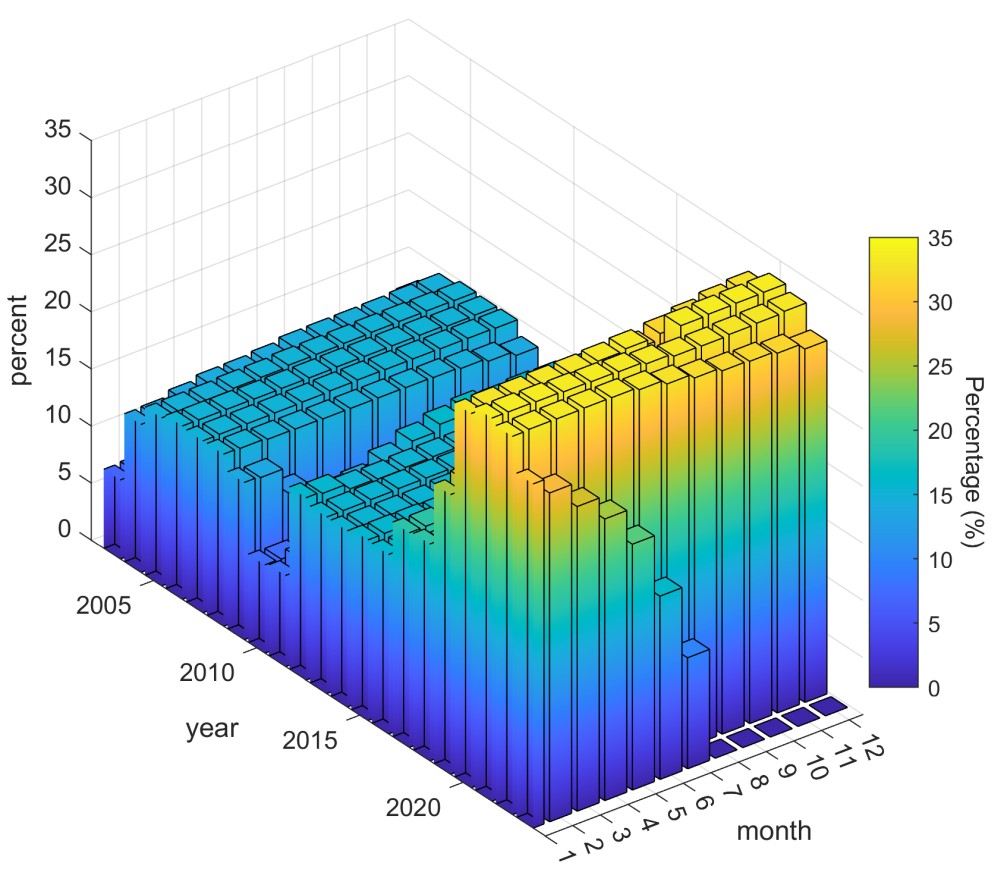

**Figure 15. Average percentage of effective raw grids for each month from 2003 to 2023.**

**3.8 Correction for elastic VLM, glacial isostatic adjustment, and firn compaction and associated uncertainties**

**3.8.1 Glacial Isostatic Adjustment**

The observed ice surface elevation changes were corrected for bedrock movement caused by elastic vertical land motion (VLM) in response to present-day mass changes and long-term past ice changes (Glacial Isostatic Adjustment, GIA). To correct for GIA, we use the GNET-GIA empirical model of Khan et al. (2016). For each grid point on a 0.5x0.5 km grid, we estimate the GIA uplift rate $dh_{GIA}$ and the associated uncertainty $\sigma_{GIA}$ retrieved from Khan et al. (2016). The GIA correction is between -3 and +16 mm/yr and the associated uncertainty, $\sigma_{GIA}$, is between 0 and 3 mm/yr (see figure 3 of Khan et al 2016). This correction of vertical land motion corresponds to total mass loss correction of 2.8±1.2 Gt/yr or 56 Gt over 20 years.

**3.8.2 Elastic vertical land motion**

We correct for elastic VLM of the bedrock by convolving monthly mass loss estimates with the Green's functions derived by Wang et al. (2012) for elastic Earth model with refined crustal structure from Crust 2.0. (Laske et al. ,2012). For each grid point, we estimate the elastic uplift rate $dh_{elas}$ and the associated uncertainty $\sigma_{elas}$. The uncertainties, $\sigma_{elas}$, are estimated by convolving uncertainties of monthly mass loss estimates with the Green's functions for elastic Earth model. The mean annual elastic correction varies between -2 to +37 mm/yr and the associated uncertainties are between 0 and 2 mm/yr. The average elastic correction over from 2003 to 2023 is 6.5±0.4 Gt/yr.

**3.8.3 Converting volume to mass**

We convert volume to mass as described in *Khan et al. (2022b)*. Conversion of the volume loss rate into the mass loss rate requires assumptions about density; therefore, using a constant ice sheet density would be inaccurate. Firn compaction must be taken into account to convert volume to mass correctly. Hence, elevation changes due to firn compaction are modeled with a simple firn model that includes melt and refreezing. It is forced by annual temperature, accumulation, melt, and refreezing from the regional climate model RACMO2.3p2 (Noël et al., 2018) at 5.5 km horizontal resolution. For each grid point, $i$, we estimate the firn compaction rate $dh_{i,firn}$ and the associated uncertainty $\sigma_{i,firn}$ as described by Khan et al 2022b. Uncertainties are estimated as described in Kuipers Munneke et al. (2015) (see their equations 8 and 9) using input fields from RACMO2.3p2. The average elastic correction over from 2003 to 2023 is 15.8±2.4 Gt/yr.

**Elevation change:**

The interpolation was performed using the ordinary kriging method (Hurkmans et al., 2014; Nielsen et al., 2013). We first used the observed annual elevation change rates to estimate an empirical semi-variogram. Next, we fit an exponential model variogram for each monthly interval with a range based on empirical semi-variogram (spanning between 40 and 90 km) to

the empirical semi-variogram to take the spatial correlation of elevation change rates into account in the error budget. For each grid point, we interpolate (using kriging) elevation change rate $dh_{i,krig}$ and the associated error $\sigma_{i,krig}$.

The total elevation change rate for each grid point $i$ is:

$$dh_i = dh_{i,krig} - dh_{i,elas} - dh_{i,GIA} - dh_{i,firn} \qquad (6)$$

Assuming the uncertainty in each of these terms is independent, we estimate the total associated uncertainty by summing each uncertainty term in quadrature:

$$\sigma_i = \sqrt{\sigma_{i,krig}^2 + \sigma_{i,elas}^2 + \sigma_{i,GIA}^2 + \sigma_{i,firn}^2} \qquad (7)$$

The total GIS mass change error for each month, $t_n$, is

$$\sigma_{month}(t_n) = \sum_{i=1}^{i_{max}} \sigma_i \qquad (8)$$

Where $i_{max}$ is the total number of grid cells that cover the GIS.

We generate a time series of cumulative GIS mass change by integrating our monthly time series of mass change over time. We estimate the cumulative errors as the root sum square of errors,


$$\sigma_{cumul}(t) = \sqrt{\sum_{n=1}^{n_{max}} \sigma_{month}^2(t_n)} \qquad (9)$$

Where $n_{max}$ is the total number of months from January 2003 to July 2023 and is set to $n_{max}=247$ months for this study.

## 4. Result

Satellite and airborne altimetry (denoted as altimetry in figure 16) indicate an ice sheet mass loss of 4,352 ± 315 Gt (12.1 ±
0.9 mm SLE) from January 2003 through August 2023, excluding peripheral glaciers (PG). Our results are similar to previous studies (Bevis et al., 2019; Mankoff et al., 2021; Sasgen et al., 2020; Velicogna et al., 2020) suggesting enhanced ice loss during the summer months of 2010, 2011, 2012, and 2019 (Figure 16 and figure 17). Ice loss was slower in 2013-2018 and was followed by an increased ice loss rate during 2020–2023.

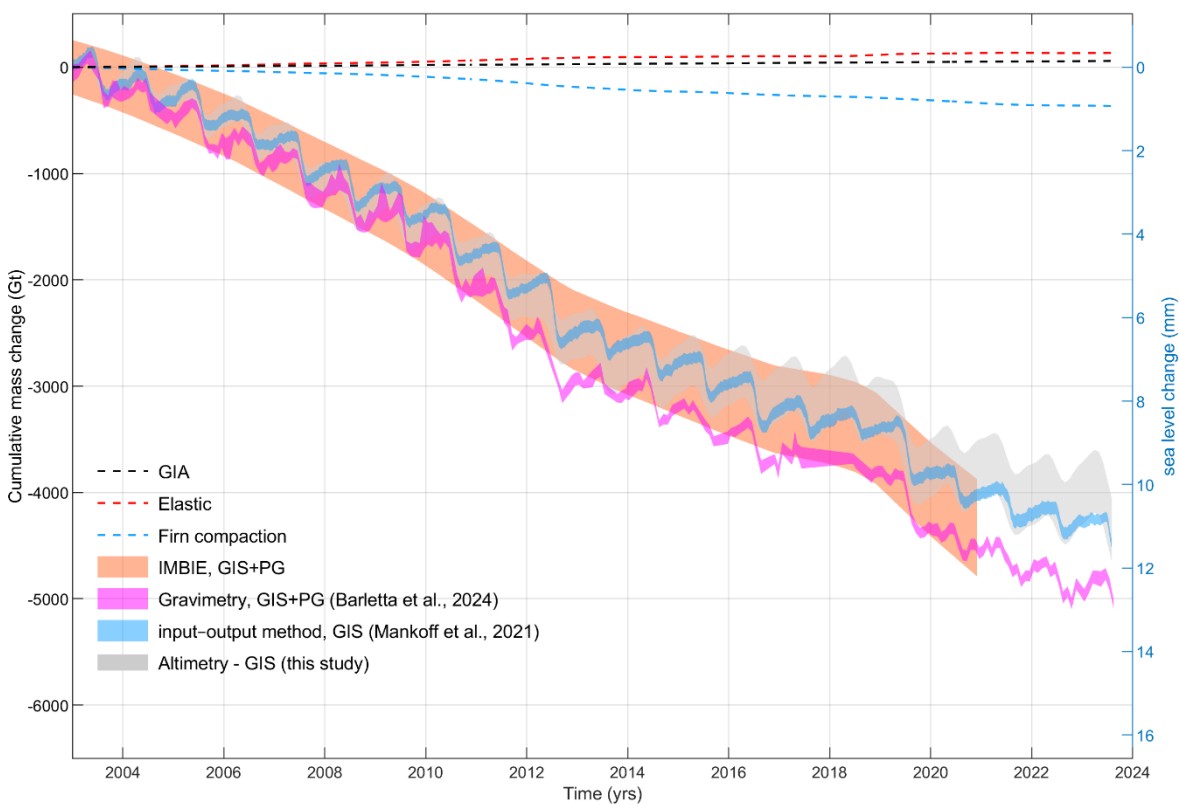

**Figure 16. Time series of cumulative monthly ice mass change of the GIS from January 2003 to August 2023 in gigatons (left axis) and sea level rise equivalent (right axis). The purple curve displays GIS+PG mass change from satellite gravimetry adopted from Barletta et al. (2020). Brown curve shows GIS+PG mass change from IMBIE. The blue curve shows GIS mass change from the Input-Output method from Mankoff et al., (2021) extended to august 2024. The grey curve shows GIS mass change from satellite and airborne altimetry from this study. The shadings represent the associated uncertainties. Corrections for GIA (black dashed**
**line), elastic deformation (red dashed line), and firn compaction (blue dashed line).**

Time series of cumulative monthly ice mass change of the GIS from January 2003 to August 2023 along with the correction of GIA and elastic induced vertical land motion and firn compaction are displayed in figure 16.

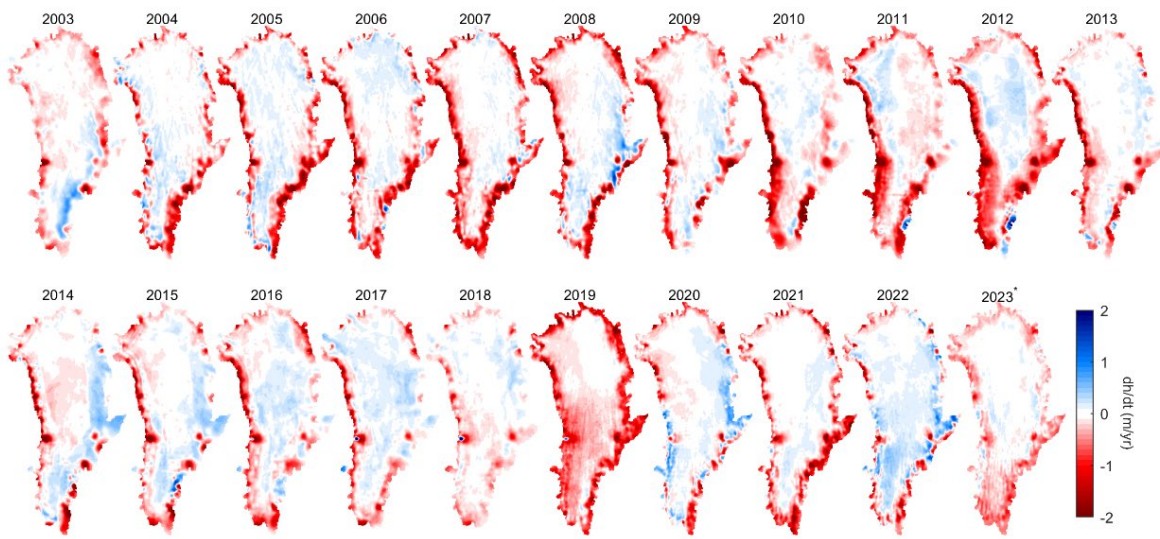

**Figure 17. Maps of annual elevation change rates from 2003 to 2023 from satellite and airborne altimetry from thus study.**


## 5. Validation

### 5.1 Mass change from satellite gravimetry and the Input-Output method

To validate our monthly Greenland ice sheet elevation changes, we use mass change from satellite gravimetry from Barlette et al., (2020) (denoted as "gravimetry"), mass change from the Input-Output method from Mankoff et al., (2021) (denoted as

"IOM"), and mass change from IMBIE (IMBIE, 2023).

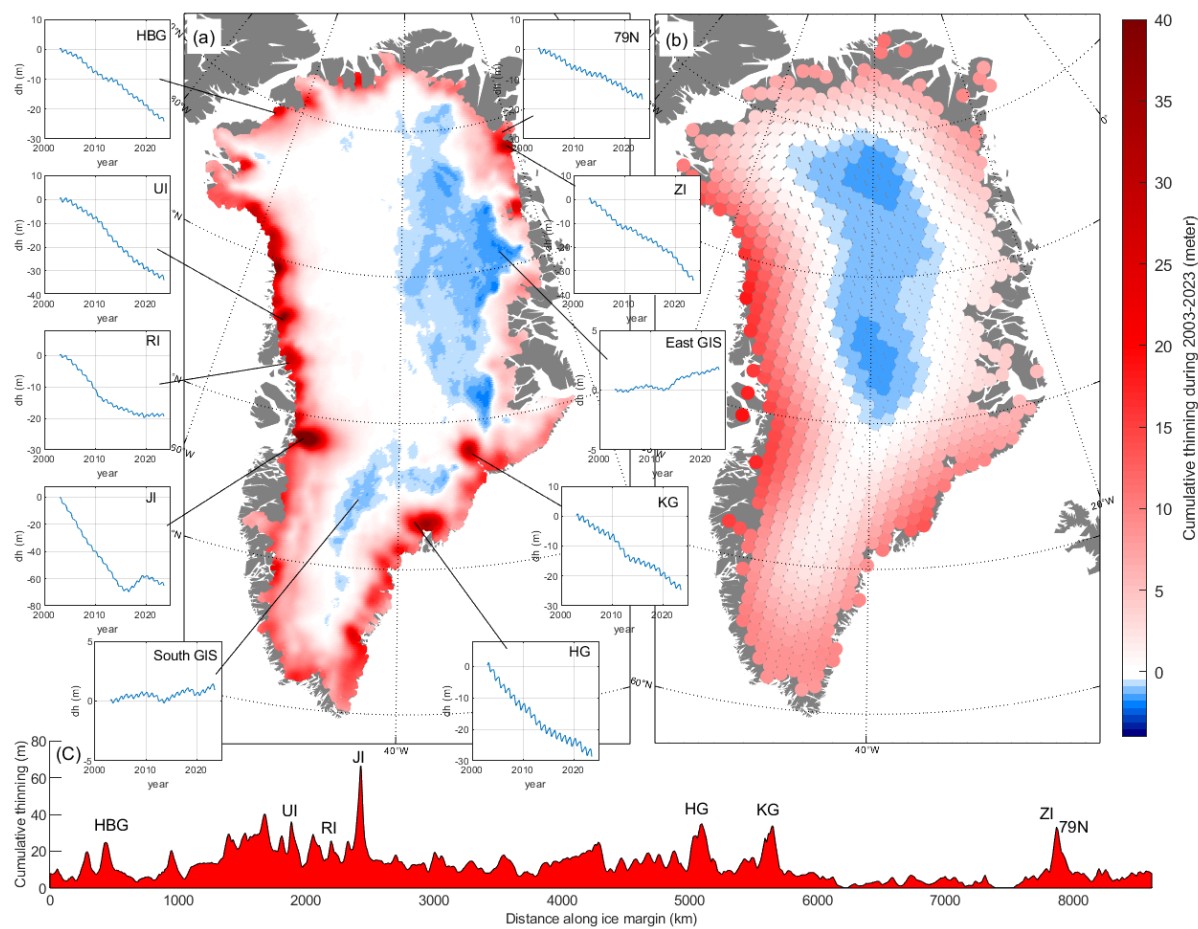

**Figure 18. (a) Cumulative surface elevation change from altimetry from January 2003 through August 2023. Sub-panels show time-series of surface elevation changes (dh) for selected locations on Helheim Glacier (HG), Kangerlussuaq Glacier (KG), east interior GIS (East GIS), Rink Isbræ (RI), Zachariæ Isstrøm (ZI), Nioghalvfjerdsfjorden (NG), Humboldt Glacier (HBG), Upernavik Isstrøm (UI), Jakobshavn Isbræ (JI), and southern interior GIS (South GIS). (b) Cumulative surface elevation change inferred from gravimetry from January 2003 through August 2023. (c) Cumulative thinning inferred from altimetry along the GIS margin during 2003–2023.**

Satellite gravimetry indicates an ice loss of 5,198 ± 173 Gt from January 2003 through August 2023, encompassing both the GIS and peripheral glaciers. The spatial resolution of satellite gravimetry, approximately ~300 km, prevents differentiation between the ice sheet and peripheral glaciers. The difference of 846 ± 359 Gt between satellite altimetry and gravimetry approximates the ice loss reported from Greenland's peripheral glaciers (Khan et al., 2022b). The cumulative surface elevation changes inferred by satellite gravimetry (Barletta et al., 2020, Barletta et al., 2024) suggest an increase in ice thickness throughout the high-level interior of Central and North Greenland (Figure 18b). This observation is inconsistent with our findings obtained from satellite altimetry measurements, which suggest that high-level thickening is limited to NE

Greenland and the saddle region between the main and south domes of the ice sheet. Satellite gravimetry does not capture the full magnitude of ice thinning around the periphery of the ice sheet due to the low spatial resolution.

The cumulative mass change from the IOM, as reported by Mankoff et al. (2021) and extended to August 2024, falls within the uncertainty range of mass change measured by altimetry (see Figure 16). Figure 16 also shows the cumulative mass change from IMBIE, which shows a larger mass change compared to altimetry due to the inclusion of peripheral glaciers. Overall, the cumulative mass changes from IMBIE, IOM, gravimetry, and altimetry over the past two decades agrees within uncertainties.

To compare multiannual ice mass change variability, we first detrend the cumulative mass changes from IOM, gravimetry, and altimetry. Figure 19a shows the three detrended time series along with the coefficient of determination, $R^2$, between altimetry and gravimetry, altimetry and IOM, and IOM and gravimetry. On multiannual timescales, there is a strong correlation between the time series, with $R^2$ values ranging from 0.88 to 0.92. Figures 19b-d display scatter plots of annual mass change rates comparing altimetry vs. gravimetry, altimetry vs. IOM, and gravimetry vs. IOM. The correlation coefficients for these comparisons range from 0.58 to 0.80, with the best correlation observed between gravimetry and IOM. This is expected, as altimetry provides smoothed elevation changes, whereas IOM and gravimetry can detect rapid changes. In general, while the three methods agree on the total mass loss of the ice sheet, there is less agreement on the precise temporal and short-term distribution of this mass loss.

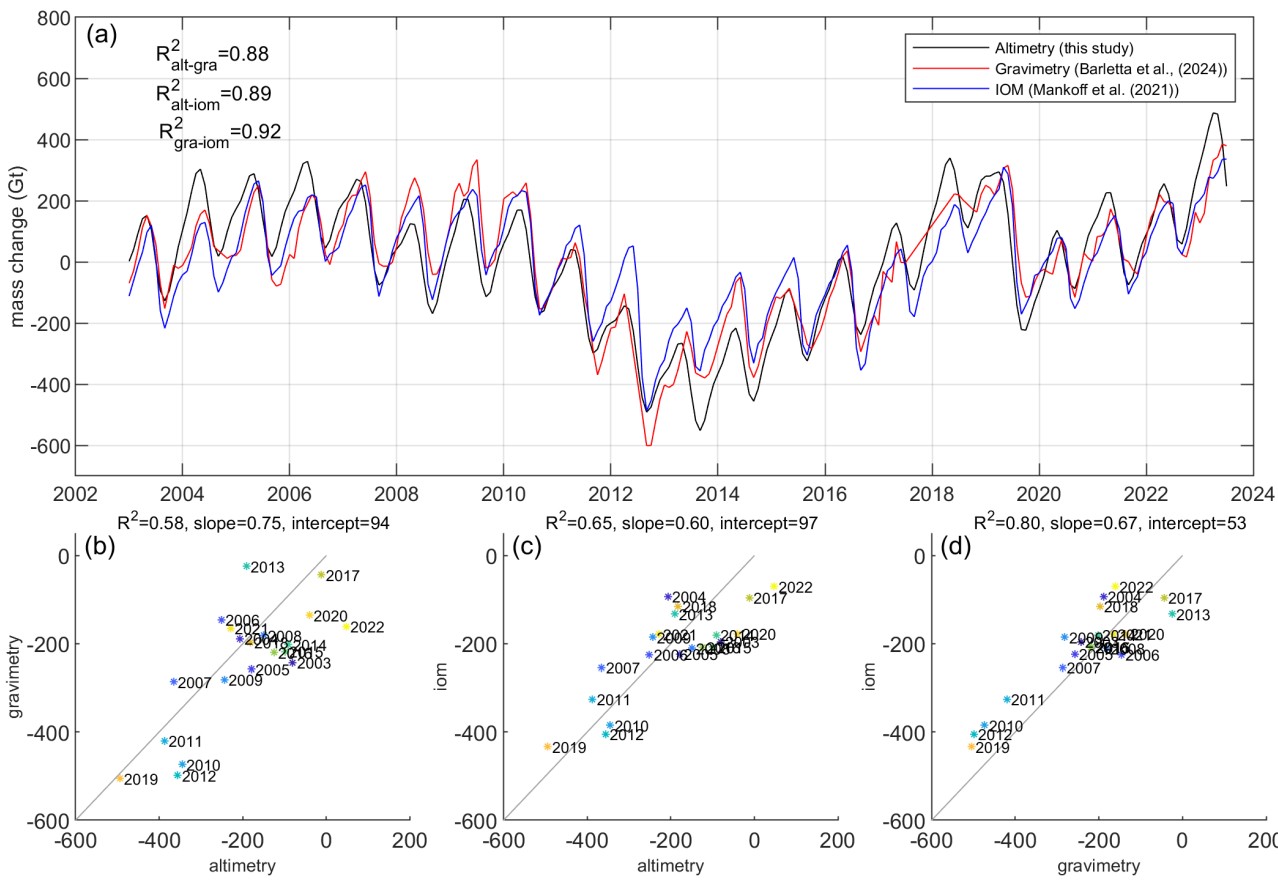

**Figure 19: (a) Detrended cumulative mass change from altimetry, gravimetry and IOM. coefficient of determination, $R^2$, between altimetry-gravimetry, altimetry-IOM and IOM-gravimetry. (b) comparison of annual mass change rates between (b) altimetry-gravimetry, (c) altimetry-IOM and (d) gravimetry-IOM. (b-d) axis units are Gt/yr.**

## 5.2 Local elevation changes

Our altimetry observations, interpolated to a 1 km grid, enable a detailed examination of GIS mass change at both regional and individual glacier scales. Figure 18 illustrates the total (cumulative) surface elevation change from January 2003 through August 2023 along with point time series at selected outlet glaciers and regions, and cumulative thinning from altimetry around the GIS margin (Figure 18c).

A large spatial and temporal variability is observed, with the entirety of West Greenland exhibiting surface lowering extending deep inland. In contrast, the surface elevation of interior Northeast Greenland increased over the past two decades. Examining glacier-specific details, our altimetry time series reveals a net thinning of ~70 m near the terminus of Jakobshavn

Isbræ. This thinning stopped and this sector underwent thickening during 2016–2018, followed by a return to thinning from 2019 to 2023, a phenomenon examined by *Khazendar et al.* (2019).

Upernavik Isstrøm (UI) has experienced a ~30 m thinning over the past two decades, with intensified thinning 2010 to 2012. Northwest Greenland's Humboldt Glacier (HBG) has thinned by 23 m over the same period, at a constant rate. One of the two major glaciers in northeast Greenland, Nioghalvfjerdsfjorden Glacier (NG), also known as 79 North, exhibits a total thinning of 15 m, while the other, Zachariae Isstrøm (ZI), shows more substantial thinning of about 35 m. In the southeast, Kangerlussuaq Glacier (KG) has thinned by approximately 23 m, while Helheim Glacier (HG) has experienced a 28 m thinning from January 2003 through August 2023. This aligns with the findings of *Williams et al.* (2021), who observed a cumulative surface elevation change of approximately 15 m during the period 2011-2020. The two time series labelled as "East interior GIS" and "South interior GIS" are located in the interior of the ice sheet (about 150 km and 250 km from the ice margin, respectively) and depict a small thickening of a few meters over the past two decades.

### 5.3 Seasonal signal

Figure 20 shows seasonal mass variability from Altimetry, GRACE, IOM, and SMB data. For each method, we plot the seasonal signal for each year on the same graph, stacking them together. To ensure consistency, we detrend the data and remove the mean for each year, setting the seasonal mass to 0 at time = 0 and 1 year. In Figure 20d, we show the seasonal signal from SMB alongside a conventional cosine function (red curve), which represents a mass increase over 6 months and a decrease over the following 6 months.

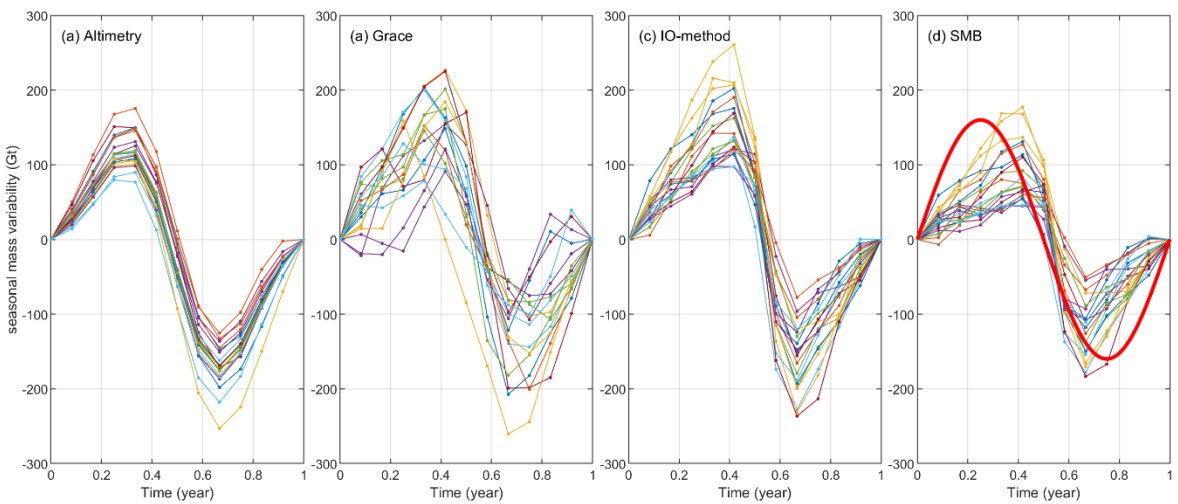

**Figure 20. Seasonal signals during 2003-2023, from (a) altimetry, (b) GRACE and GRACE Follow-On, (c) IOM, and (d) SMB. Each curve represents seasonal signal from January to December. The red curve in (d) displays conventional cosine function.**

Our results show that GRACE and IOM, which are based on direct mass change observations, align better with our seasonal model than a conventional cosine function. While the cosine function commonly used in many studies provides a useful

first-order approximation for describing the seasonal signal, our model is more consistent with the observed data. The SMB model incorporates accumulation, runoff, and evaporation processes. Previous studies have highlighted the correlation between accumulation and melting with the North Atlantic Oscillation (*Bevis et al.*, 2019; *Bjørk et al.*, 2018; *Noël et al.*, 2018).

**5.4 Temporal agreement between approaches during the rapid ice**

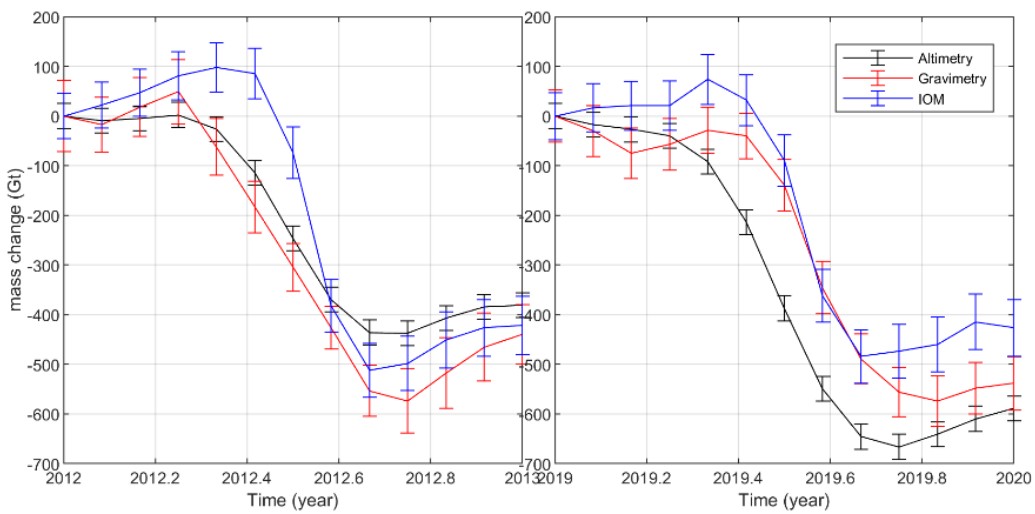

**Figure 21. Rapid ice loss in 2012 and 2019 observed by mass loss during satellite altimetry, GRACE, and the input-output method.**

In 2012 and 2019, the Greenland Ice Sheet experienced record-high ice loss during the summer months, as observed by

satellite altimetry, GRACE, and the input-output method. In both years, extreme melt events were driven by anomalously warm atmospheric conditions, leading to significant surface mass loss (Bevis et al., 2019). Satellite altimetry recorded a rapid decline in ice surface elevation, while GRACE data detected substantial reductions in gravitational mass, confirming extensive ice loss. The input-output method further confirmed the ice mass loss. Figure 21 illustrates the temporal level of agreement between the three approaches during the rapid ice losses in 2012 and 2019. All three methods detected ice loss

ranging from 381 to 439 Gt in 2012 and 426 to 589 Gt in 2019.

## 6. Discussion

The observed spatial and temporal resolution of ice surface elevation changes from satellite altimetry is of critical importance for better quantifying the processes impacting ice sheet change, and for advancing and refining ice sheet modelling. This level of precision is crucial for accurately representing complex processes within ice sheet models, such as seasonal ice flow dynamics, and changes in surface mass balance *(Goelzer et al.,* 2017). Recent advances in machine learning and automatic differentiation tools enable transient calibration in ice sheet modelling, facilitating better estimation of unmeasurable parameters such as the basal sliding coefficient and ice rheology. The integration of fine-scale data into models enhances their ability to simulate realistic responses to climate variables, contributing to more accurate predictions of ice sheet behaviour and potential impacts on sea-level rise. Additionally, high-resolution satellite altimetry data helps identify local trends and patterns, allowing for a more nuanced understanding of regional variations in ice sheet dynamics *(Mankoff et al.,* 2019; *Mankoff et al.,* 2021). Recent studies using high-resolution modeling of Greenland's major outlet glaciers has shown that short-term changes in terminus position, ice thickness, and basal conditions significantly influence ice velocity *(Cheng et al.,* 2022; *Lippert et al.,* 2024, *Lu et al.,* 2025). For example, studies on Helheim Glacier (100–1,500 m resolution), Kangerlussuaq Glacier (350 m–12 km), and Jakobshavn Isbræ (100–1,500 m) have all demonstrated that ice front retreat and thickness variations drive substantial seasonal and multi-annual ice velocity fluctuation. These studies emphasize that annual elevation changes at a 5 km or higher resolution risk averaging out critical seasonal dynamics, leading to inaccuracies in modeling ice dynamics and underestimating short-term variations that are essential for projecting future changes of the ice sheet. Ultimately, the incorporation of observed high-resolution data into ice sheet models is essential for improving the fidelity of simulations and enhancing our ability to assess the implications of climate change on ice sheet stability and sea-level rise *(Choi et al.,* 2023).

In addition, a 1×1 km grid resolution of ice surface elevation data is essential for accurately modeling elastic land deformation of the crust because it captures the spatial variability of ice load changes at a fine enough scale to resolve localized flexural responses. Ice mass variations exert pressure on the Earth's crust, causing it to deform elastically, but these deformations are not uniform across the ice sheet. In regions with steep ice surface gradients, such as outlet glaciers and ice sheet margins, coarse-resolution data may smooth out critical variations in ice load, leading to inaccuracies in predicted uplift and subsidence patterns (Khan et al., 2022). A high-resolution grid allows for more precise calculations of surface mass redistribution, improving estimates of bedrock displacement. This level of detail is particularly crucial when observing Glacial Isostatic Adjustment with GPS observations, where corrections for elastic deformation need to be applied.

Combining data from sensors with different spatial footprints presents challenges in accurately capturing small-scale elevation changes. Sensors with coarse spatial resolution tend to smooth out localized ice surface variations, potentially underestimating rapid or heterogeneous changes. In contrast, higher-resolution sensors provide more detail but often have

limited coverage or increased noise. Merging datasets requires careful interpolation to reconcile differences in sampling density, measurement techniques, and error characteristics. Discrepancies in spatial footprints can also result in mismatches when detecting localized thinning, particularly at glacier termini or steep ice sheet margins, which may affect estimates of mass loss and ice dynamics at finer scales.

A key limitation in detecting rapid ice sheet elevation changes using satellite altimetry is the temporal resolution of the data. Many altimetry satellites have repeat cycles spanning months, making it difficult to capture short-lived or sudden elevation changes, such as those driven by extreme melt events or rapid ice flow acceleration. Gaps between observations can lead to underestimation or misinterpretation of transient changes, especially in highly dynamic regions where ice loss occurs on short timescales. Additionally, seasonal variations in surface conditions, such as snowfall accumulation or meltwater refreezing, introduce further uncertainties when interpolating between measurement periods.

Since most ice loss occurs at the ice sheet margin, where the terrain is rough and data coverage is sparse, an alternative approach may be necessary. One method involves fitting a third-order polynomial equation to describe the surface shape using observations within a 1 km radius. While this approach works well for much of the ice sheet, it may be insufficient in fast-flowing regions with rugged terrain. Using a higher-order polynomial is not feasible due to the limited number of observations relative to the unknown parameters in Equation 1. Additionally, we assume that surface topography remains constant over time intervals of 4–7 years. While this is a reasonable approximation for most of the ice sheet, near the termini of outlet glaciers, topography can change significantly from year to year. To address these challenges, integrating high-resolution (10×10 m) annual Digital Elevation Model (DEM) data with altimetry observations may improve topographic representation (Winstrup et al., 2024).

## 7. Limitations of the method and data

**Temporal Resolution:** We estimate monthly elevation changes by fitting a 7th-order polynomial and a seasonal signal to the observed elevation time series. This approach, which we refer to as "smoothed" elevation changes, results in smoothed data that cannot detect rapid elevation changes. The method's effectiveness is limited by the temporal resolution of available airborne and satellite altimetry data. For instance, ICESat provides 2-4 repeat measurements per year at the same location, while ATM data offers only one measurement annually. This means that measurements should not be expected to resolve at high spatial resolution for ice-sheet changes at sub-seasonal time scales, only smoothed sub annual changes using some pre-defined seasonal model are possible.

**Spatial Resolution:** The spatial resolution of elevation changes estimated from satellite radar altimetry over ice sheet margins is constrained by the coarse spatial footprint of instruments like the EnviSat radar altimeters, leading to a lack of fine-scale detail. The steep and rugged topography of ice sheet margins causes signal scattering and reflection issues, resulting in inaccurate measurements. Figures 2 and 3a illustrate the poor spatial resolution along the ice margin, especially

for EnviSat. Another limitation is that NASA's ATM flights cover main outlet glaciers but do not survey all glaciers annually. Generally, NASA's ATM flight coverage is denser in West Greenland than in East Greenland. Additionally, the Operation IceBridge campaign, with measurements conducted once per year, does not allow for the detection of seasonal signals. However, the ICESat and ICESat-2 missions significantly enhance our understanding of ice sheet elevation changes, including in the ice margin regions and seasonal signals.

In general, our method lacks detecting rapid ice elevation changes. For example, drainage of surface lakes on the Greenland Ice Sheet are not detected. A small area with elevation changes of up to tens of meters will be considered as an outlier. We detect local outliers of monthly rates, using planar regression in 20 km bins. Drainage of surface lakes, which often occurs over a small area will be detected as an outlier using our method. Also, Instances of rapid elevation changes occurring over extremely short time spans will go unnoticed through our approach. For instance, rapid accelerations and decelerations, as illustrated by *Vijay et al.* (2021), taking place within a 2-month period are not fully captured by our surface elevation change product. Nevertheless, our method effectively identifies melt-driven rapid thinning episodes during the warm summers of 2010-2012 and 2019. Furthermore, in Greenland, 87% of the glacierized region terminates in the ocean leading to frontal ablation, involving both ice discharge and terminus retreat. We note that our mass loss estimate from satellite altimetry does not incorporate mass loss below sea level, constituting less than 10-20% of the overall frontal ablation, as indicated by *Kochtitzky et al.* (2023) and *Greene et al.* (2024).

The kriging interpolator's weights are determined by the modeled variogram, making it highly sensitive to any mis-specification of the variogram model. Its interpolation accuracy is limited when the number of sampled observations is small, the data has a restricted spatial extent, or there is insufficient spatial correlation. In such cases, constructing a reliable sample variogram becomes challenging. Using data from a single sensor—such as CryoSat-2 or EnviSat—near the ice margin (see Fig. 8g and 8h) where data gaps are large can lead to significant large uncertainty. However, our approach, which integrates multiple data sources, particularly the inclusion of ATM data concentrated along glacier flow lines, helps to reduce uncertainty. However, ATM data does not provide complete coverage of all glaciers in Greenland. In particular, elevation changes in small glaciers, especially those 1–2 km wide in southeast Greenland, may not be well captured.

## 8. Data products and availability

Our main products are gridded time series of monthly Digital Elevation Models and gridded monthly time series of surface elevation change (in water equivalent). However, we also deliver surface elevation change (in ice equivalent). This allows users to convert ice to mass using their own model for e.g. firn compaction. To make the data user friendly for the ice sheet modelling community, we deliver code that can create time series of elevations with respect to geoid model (often needed as input in numerical ice flow model).

To make our elevation change products useful for the ice sheet modeling community, we deliver a Digital Elevation Model (DEM). We use the DEM from the Greenland Ice Mapping Project (GIMP) (Howat et al., 2015), downscaled to a 1x1 km

grid resolution to align with the elevation change products developed in this study. Additionally, we convert it to a reference time of January 1, 2003, using the dh/dt product developed in this study.

**Table 2: Data products on a 1x1 km grid covering the Greenland Ice sheet.**

| Data product | Spatial Resolution | Temporal Resolution | Temporal Coverage |
|---|---|---|---|
| Digital Elevation Models | 1x1 km | One epoch* | 2003.00 |
| Monthly Elevation changes (in water equivalent) | 1x1 km | Monthly | 2003 to 2023 |
| Monthly Elevation changes (in ice equivalent) | 1x1 km | Monthly | 2003 to 2023 |
| Elastic VLM | 1x1 km | Monthly | 2003 to 2023 |
| Firn compaction rate | 1x1 km | Monthly | 2003 to 2023 |
| GIA | 1x1 km | One epoch* | 2003.00 |
| Geoid model | 1x1 km | One epoch** | 2003.00 |

\*) we assume vertical land motions due to GIA does not change over a 20 yrs period.

\*\*) we assume the geoid does not change over a 20 yrs period.

All grid files use WGS 84 / NSIDC Sea Ice Polar Stereographic North (EPSG:3413)

Our main products are Digital Elevation Model and gridded monthly time series of surface elevation change (in water equivalent). However, we also deliver surface elevation change (in ice equivalent). This allows users to convert ice to mass using their own model for e.g. firn compaction or GIA. To make the data user friendly for the ice sheet modelling

community, we deliver geoid model (often needed as input in numerical ice flow model).

We provide a code snippet to estimate the parameters for equations 2 to 4 using the test input data.

**File naming and data format.**

We provide 2 zip files.

File 1: Greenland_geotiff_1kmgrid.zip

File 2: Greenland_netcdf_1kmgrid.zip

file 1 format: NetCDF (monthly dh/dt data)

file 2 format: geotiff (mean dh/dt from 2003-2023)

File 1 content: This is a zip file that contains the following netcdf files.

- **Greenland_DEM_1kmgrid.nc** (Digital Elevation Models)
- **Greenland_dhdt_elas_1kmgrid.nc** (monthly elastic Vertical Land Motion)
- **Greenland_dhdt_elas_err_1kmgrid.nc** (error associated with monthly elastic Vertical Land Motion)
- **Greenland_dhdt_firn_1kmgrid.nc** (monthly firn compaction rate)
- **Greenland_dhdt_firn_err_1kmgrid.nc** (error associated with monthly firn compaction rate)
- **Greenland_dhdt_GIA_1kmgrid.nc** (Glacial Isostatic Uplift rates)
- **Greenland_dhdt_GIA_err_1kmgrid.nc** (error associated with Glacial Isostatic Uplift rate)
- **Greenland_dhdt_icevol_1kmgrid.nc** (Monthly Elevation changes (in ice equivalent))
- **Greenland_dhdt_icevol_err_1kmgrid.nc** (error associated Monthly Elevation changes (in ice equivalent))
- **Greenland_dhdt_mass_1kmgrid.nc** (Monthly Elevation changes (in water equivalent))
- **Greenland_dhdt_mass_err_1kmgrid.nc** (error associated with Monthly Elevation changes (in water equivalent))
- **Greenland_geoid_1kmgrid.nc** (geoid height)

Each file contain a header with information about data types.

File 2 content: same as file 1 but with geotiff format.

## 9. Conclusion

The spatial and temporal resolution of elevation change products is limited by the resolution of satellite ground tracks. Satellite altimetry usually offers 2-5 repeat measurements per year over the same location. To achieve higher temporal resolution (e.g., 1 month), elevation changes must be averaged over a large area, which leads to coarse spatial resolution. Alternatively, we model a seasonal signal (see Methods) and generate "smoothed" monthly elevation changes with higher spatial and temporal resolution. This allows us to produce an elevation change product with detailed insights into the
dynamics of the Greenland Ice Sheet from January 2003 to August 2023. Our product reveals a cumulative ice loss of 4,352 ± 315 Gt, corresponding to a global mean sea level rise of 12.1 ± 0.9 mm over two decades and local thinning of up to 70 m near the terminus of JI. Validation of our monthly mass changes of the Greenland ice sheet, against mass change from satellite gravimetry and the Input-Output method, suggests strong correlation with R values ranging from 0.88 to 0.92. Incorporating our new high-resolution data into ice sheet models can enhance our understanding of ice dynamics, thus
improving predictions about the Greenland Ice Sheet's response to climate change and its impact on sea-level rise. Despite certain limitations, especially in capturing short, rapid elevation changes, our methodology offers a valuable tool for monitoring and analyzing the evolving dynamics of the Greenland Ice Sheet.


**Author contributions.**

SAK conceived the study, analysed most of the data. VH processed Cryosat-2 data. VB analysed GRACE abd GRACE-FO
data. KDM provided and prepared IOM data. Writing by SAK, HS, MM, WC, GC, DB, VRB, NKL, WK, MvdB, KHK, AA, BN, JEB, JAM, RSF, KDM, IMH, KO, DF, AL, EYHL, and JH. Project administration by SAK, WC, HS and NKL. IMH developed DEM model.

**Competing interests.**

Author Kenneth D. Mankoff is a member of the editorial board of the journal.

**Acknowledgements**

S A Khan, H Seroussi, M Morlighem, W Colgan, G Cheng, K Kjær, N K Larsen, and A Løkkegaard acknowledge support from the NOVO Nordisk foundation grant no NNF23OC00807040. This is a publication of the Center for Ice Sheet and Sea
Level Predictions (CISP). B Noël was funded by the Fonds de la Recherche Scientifique de Belgique (F.R.S.-FNRS). R.S Fausto and J.E. Box acknowledges support from the Programme for monitoring of the Greenland ice sheet (PROMICE). K. Mankoff was supported by the NASA Modeling Analysis and Prediction program.

**Data and code Availability**

Monthly elevation change rates of the GIS from 2003 to 2023 is available at the following data repository https://datadryad.org/stash/share/RFbPIGTZRn0Vs0u8hj1PSgjde11IPAL-k8TvEOOEBDA (Khan et al., 2024a). The DTU GRACE and GRACE-FO Greenland Mass Balance product is available at https://doi.org/10.11583/DTU.12866579.v4 (Barletta et al., 2020). Code is available at https://doi.org/10.5281/zenodo.13276108 (Khan et al., 2024b)

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
