# Peer review of "Smoothed monthly Greenland ice sheet elevation changes during 2003-2023"

_Earth System Science Data, 2024_

## Referee Comment (RC2)

**Review report**

**Manuscript Number/DOI:** https://doi.org/10.5194/essd-2024-348

**Full Title**: Smoothed monthly Greenland ice sheet elevation changes during 2003-2023

**Authors:** Shfaqat A. Khan et al.

Submitted to *Earth System Science Data*

**Recommendation:** Major/Moderate revision

**Overall Evaluation**

This manuscript presents a comprehensive analysis of Greenland Ice Sheet elevation changes from 2003 to 2023, integrating multiple satellite and airborne altimetry datasets. The methodology demonstrates considerable rigor in data processing and uncertainty assessment, particularly in combining diverse data sources to produce a consistent long-term record. The approach to data integration and uncertainty quantification shows careful attention to detail. However, several aspects of the analysis require additional clarification and enhancement to strengthen the scientific contribution of this work. These concerns primarily relate to the physical basis of the seasonal model, methodology justification, and validation approaches.

**Major Scientific Concerns and Suggested Improvements**

*Seasonal Signal Modeling and Physical Basis*

The seasonal signal modeling presented in Section 3.2 (pages 7-8) requires substantial revision. The authors propose a new seasonal model in equation (4) that assumes 8 months of mass gain and 4 months of mass loss. While Figure 5 illustrates this seasonal pattern, the physical basis for this temporal distribution needs more rigorous justification. Specifically, the manuscript should explain how this seasonal pattern relates to known atmospheric circulation patterns and seasonal precipitation variability across Greenland. The relationship with regional climate dynamics, including the influence of the North Atlantic Oscillation on seasonal mass balance patterns, should be addressed. The authors should also demonstrate why their model performs better than the conventional cosine function described in equation (5), particularly in capturing the asymmetric nature of accumulation and ablation processes.

*Methodology and Parameter Selection*

The methodology section (Section 3.2-3.4) should better justify key analytical choices. The use of a 7th-order polynomial for fitting elevation changes (equation 2, page 7) lacks sufficient justification. The authors should demonstrate why this order is optimal by comparing residuals across different polynomial orders and discussing potential overfitting issues. A systematic analysis of model performance with different polynomial orders would strengthen this choice. Additionally, the kriging interpolation parameters described on page 17 (lines 334-335) need more detailed explanation, particularly regarding the choice of the 65 km range parameter. The spatial correlation structure of elevation changes and its influence on interpolation parameters should be more thoroughly discussed.

*Validation and Comparison*

The validation approach presented in Section 5 (pages 20-23) should be expanded. While the comparison with GRACE data and the Input-Output method provides valuable insight, the analysis should include:

- Quantitative metrics for agreement between different methods, including correlation coefficients and root-mean-square differences
- Analysis of spatial patterns in the differences between methods, particularly in regions with complex topography
- Discussion of temporal variations in the agreement between different approaches, especially during periods of rapid change
- Assessment of seasonal cycle differences between methods and their implications for mass balance estimates

*Discussion and Implications*

The discussion section (Section 6, pages 24-25) should be expanded to address methodological limitations more comprehensively. The authors should discuss:

- The implications of combining data from sensors with different spatial footprints, particularly for capturing small-scale elevation changes
- The challenges in detecting rapid elevation changes and their impact on mass balance estimates
- The potential impact of these limitations on ice sheet modeling applications, especially for initialization and validation
- Future improvements that could address current limitations, including upcoming satellite missions and methodological advances
- The broader implications for understanding ice sheet response to climate change

**Technical Corrections and Presentation**

*Figures and Visualization*

Several figures require improvement:

- Figure 5 (page 8): Add more detailed axis labels and improve legend readability, and if possible, include error bounds on the seasonal signals to better represent uncertainty in the temporal patterns
- Figures 13-14 (pages 20-21): Consider adding difference maps to better illustrate spatial patterns and include quantitative measures of uncertainty in the spatial comparisons

**Recommendation**

Major/Moderate Revision. The manuscript requires substantial revisions before it can be considered for publication. The authors should:

1. Provide a thorough physical justification for their seasonal model, including regional analysis and

comparison with known climate patterns

2. Strengthen the methodology section with quantitative justification for key parameter choices
3. Expand the validation analysis with comprehensive statistical metrics and spatial comparisons
4. Enhance the discussion of limitations and implications

These revisions are essential to ensure that this valuable dataset can be effectively utilized by the broader scientific community. Upon addressing these concerns, this work will make a significant contribution to our understanding of Greenland Ice Sheet mass changes and provide an important resource for future research in glaciology and climate science.

---

## Community Comment (CC1)

I'm curious what mask was used for both interior holes (nunatuks?) and the ice sheet boundary, and/or if it can be changed to address boundary issues.

Below are two images of the data product for essd 2024-311 and two images of the data product for essd 2024-348. This comment is submitted to both papers.

Interior holes can be filled (at possibly low quality) by interpolation. Missing data at the edges is harder to extrapolate.

This issue came up today after discussions with BedMachine about updates to that product. Updates need the mask and DEM to match temporally. The mask used here does not likely represent the best 'true' ice sheet mask – that would be its own product and needs to be evaluated separately. In fact none of the words 'mask', 'outline', and 'boundary' exist in 2024-348, while 2024-311 mentions using the Zwally (2012) ice sheet outline and RGI 7 for peripheral glaciers.

The Zwally mask does not capture the true edge in many places. Can this product be regenerated with a larger mask? Zwally but buffered by a few grid cells? The superset of Zwally, Mouginot, and BedMachine? What's the downside of having some land cells, or partial land cells, included in this, other than some locations where there should be no change?

If key grid cells are missing and cannot be reasonably interpolated or extrapolated, then this product cannot be used as a basis for an updated BedMachine DEM, and another product that does provide full coverage would need to be found and used instead.

There is an ongoing community effort to avoid under- or double-counting cells where RGI and ice sheet communities may overlap. Gaps in these products are counter productive to this effort.

Figures follow. Blue is BedMachine. Red outline is RGI. Orange is a 2022 remote-sensing 'best true' outline.

[Figure]

2024-311 Figure 2

2024-348 Fig 1

2024-348 Fig 2

---

## Author Response (AR1)

**Reviewer 1 comments and our response**

This manuscript presents monthly GrIS elevation changes from a long-term series using multisources satellite and airborne altimeter data. The authors improved the previous annual elevation change method to detect monthly elevation changes. They also separated the seasonal surface variation from the time series of surface elevation observations. This method seems to be effective; however, I still have some concerns about this paper.

**Authors:** Thank you very much for your time and for reviewing this paper. In light of the insightful feedback from you and the other reviewers, we have made several changes that have greatly improved the revised manuscript. A detailed response to your comments, addressing all the identified issues, is listed below.

Major comments:

1. This paper resembles more a technical report than a scientific paper because it lacks careful organization of original data and a logical description of the methods.

Authors: We follow the guidelines from ESSD regarding the organization of the paper. We agree that the paper provides no new science and is structured more like a technical report.

2. The authors use the seasonal terms derived from ICESat or ICESat-2 to represent the seasonal surface elevation changes observed in other satellite altimeters. The rationale and the associated uncertainties should be discussed further.

Authors: correct, the rationale behind this selection is Figure 12 and Figure 13.

While ICESat and ICESat-2-derived seasonal amplitude maps show the same spatial pattern (Fig. 12a and 12b), CryoSat-2-derived seasonal amplitude maps show some differences, likely caused by radar signals penetrating through surface snowfall.

Figure 8 shows the seasonal signal from CryoSat-2 and ICESat/ICESat-2 for the exact same location. We note that CryoSat-2 shows a smaller amplitude than ICESat/ICESat-2, likely due to radar signal penetration through surface snowfall.

In addition, ICESat (2003–2009) and ICESat-2 (2018–2023) show almost the same spatial pattern of the amplitude (new Figure 13).

However, the multiannual variations in surface elevations from ICESat-2 and CryoSat-2 are consistent. Similar findings have recently been published by Ravinder et al. (2024); see their Figure 2b.

 The validation and cross-comparison with other monthly GrIS elevation change methods should be discussed, such as the method developed by Lai et al. R. Lai and L. Wang, Monthly Surface Elevation Changes of the Greenland Ice Sheet From ICESat-1, CryoSat-2, and ICESat-2 Altimetry Missions, IEEE Geoscience and Remote Sensing Letters, vol. 19, pp. 1-5, 2022, doi: 10.1109/LGRS.2021.3058956

Authors: Thanks for this paper. We mention the method from Lai et al. in the introduction. In addition, we also list the recent paper Ravinder, N., Shepherd, A., Otosaka, I., Slater, T., Muir, A.,

& Gilbert, L. (2024). Greenland Ice Sheet elevation change from CryoSat-2 and ICESat-2. *Geophysical Research Letters*, 51, e2024GL110822. https://doi.org/10.1029/2024GL110822

4. The accuracy of the time series elevation change detection method depends on the validity of observations in a specific grid. With higher resolution grids, there are fewer observations. Did the authors analyze the distribution of valid observations at 1 km resolution across the whole GrIS on a monthly scale? If so, please add this distribution.

Authors: Yes, we did. We have now added a figure that shows the number of observations used in each single-point time series from each sensor.

Note that for our method, it is not necessary to consider the distribution of valid observations on a monthly scale. However, the distribution of valid observations over the entire length of the dataset is relevant. This is shown, for example, in Figure 6, which presents a surface elevation change time series derived from ICESat data for a single point. The error bars denote observed elevations, but the best-fitting 7th-order polynomial is used to estimate monthly elevation changes for this particular point.

The number of observations used in each single-point time series is shown in the figure below.